# Path-Enhanced Contrastive Learning for Recommendation

**Haoran Sun**[†]
Beijing Jiaotong University
haoran.sun@bjtu.edu.cn

**Fei Xiong**[†*]
Beijing Jiaotong University
xiongf@bjtu.edu.cn

**Yuanzhe Hu**[†]
Institute of Software Chinese Academy of Sciences
yuanzhe@iscas.ac.cn

**Liang Wang**
Northwestern Polytechnical University
liangwang0123@gmail.com

## Abstract

Collaborative filtering (CF) methods are now facing the challenge of data sparsity in recommender systems. In order to reduce the effect of data sparsity, researchers proposed contrastive learning methods to extract self-supervised signals from raw data. Contrastive learning methods address this problem by graph augmentation and maximizing the consistency of node representations between different augmented graphs. However, these methods tends to unintentionally distance the target node from its path nodes on the interaction path, thus limiting its effectiveness. In this regard, we propose a solution that uses paths as samples in the contrastive loss function. In order to obtain the path samples, we design a path sampling method. In addition to the contrast of the relationship between the target node and the nodes within the path (intra-path contrast), we also designed a method of contrasting the relationship between the paths (inter-path contrast) to better pull the target node and its path nodes closer to each other. We use Simplifying and Powering Graph Convolution Network (LightGCN) as the basis and combine with a new path-enhanced graph approach proposed for graph augmentation. It effectively improves the performance of recommendation models. Our proposed Path Enhanced Contrastive Loss (PECL) model replaces the common contrastive loss function with our novel loss function, showing significant performance improvement. Experiments on three real-world datasets demonstrate the effectiveness of our model.

## 1 Introduction

In the era of information explosion, recommender systems play a crucial role in identifying users' preferences and delivering personalized experiences effectively[18]. Among the various techniques, CF[9, 21] has become a cornerstone approach to generate recommendations utilizing implicit feedback such as clicks, purchases, and comments. The core idea of CF methods is that users with similar behaviors are likely to share similar preferences. CF methods are broadly classified into memory-based[15, 4, 20] and model-based approaches[17, 10, 12]. Recent research trends focus on model-based CF techniques due to their superior performance and scalability. However, CF methods are often challenged by the data sparsity problem. To address this, researchers proposed various models to enhance the representations of users and items by using additional information[10, 25, 2, 13, 27, 5]. For instance, models like SVD++ incorporate implicit feedback from user-item interactions to refine

---

[*]Corresponding author
[†]These authors contributed equally to this research.

predictions[12], while LightGCN effectively captures higher-order collaborative signals to improve the embedding quality of users and items[10].

Recently, contrastive learning has emerged as a powerful paradigm in representation learning, achieving remarkable success in various domains such as computer vision[3, 7, 1, 8] and natural language processing[6, 28, 31]. By leveraging self-supervised signals, contrastive learning can effectively extract meaningful features from large-scale unlabeled data, offering a promising solution to the problem of data sparsity[24, 23]. Given its ability to provide additional supervisory signals, an increasing number of studies[14, 26, 33] have applied contrastive learning techniques to recommender systems, resulting in significant improvements in recommendation performance. The core concept behind contrastive learning is creating additional supervised instances and applying a self-designed task to this augmented data, thereby tackling the challenge of data scarcity. Specifically, we randomly select a node from the interaction graph to be the target node and paths extending from the target node as center paths. Data augmentation is achieved by perturbing the interaction graph. During training, the augmentation of the target node is represented as its positive samples and other nodes as negative samples. However, disturbing interaction graph may generate irrelevant connections or discard critical training information, thereby weakening the reliability and overall performance of contrastive learning models. Moreover, treating other nodes as negative nodes will unintentionally distance the target node from its node neighbors along the interaction path. However, these neighbors exert a positive influence on the target node.

To overcome these challenges, we propose a path-enhanced contrastive learning method (PECL) that focuses on path-level representations, offering a neighborhood-based perspective for contrastive learning approaches. We first propose an intra-path contrastive learning strategy that effectively selects nodes for contrastive learning, so that the target node is pulled closer to the nodes along the interaction path. Nevertheless, the semantics of a single node can often be diverse or ambiguous, which motivates us to further design an inter-path contrastive learning method. Since a path consists of multiple sequentially connected nodes, it naturally encodes contextual constraints and thus helps to mitigate noise. To enable inter-path contrast, we devise a general path sampling strategy that selects representative paths extending from the target node as positive samples of the center path, thereby enabling the model to align semantically similar paths more effectively.

In summary, our main contributions are summarized as follows:

- We propose an efficient path contrastive learning model that utilizes multiple positive path samples to guide the updating of the representation of the center path. The experiments show that multiple positive samples together influence the representation of the center path and enhance the recommendation performance of the model.

- We design a path sampling method that can sample paths similar to the center path as positive samples. The positive samples provided allow model to perform inter-path contrastive learning, which in turn brings the target node closer to its collaborative nodes on other paths.

- By utilizing three datasets derived from actual real-world situations, we conducted empirical research that revealed a distinct advantage of PECL compared to state-of-the-art baseline models.

## 2 Related Work

**Graph Neural Network Based Recommender Systems.** Graph Neural Networks (GNNs) have emerged as a powerful tool for recommendation tasks, particularly because they can capture intricate relationships between users and items in a graph-based setting. Traditional collaborative filtering (CF) models, such as matrix factorization, have been widely adopted in recommender systems. However, these models often struggle with data sparsity and fail to capture complex higher-order interactions between users and items. GNN-based models, such as NGCF (Neural Graph Collaborative Filtering)[25], address these limitations by leveraging message-passing mechanisms to propagate information across the user-item graph. NGCF captures both direct and indirect collaborative signals, making it particularly suitable for recommendation tasks in sparse data scenarios. Despite the improvements offered by NGCF, its heavy reliance on deep message-passing layers and non-linear activations leads to potential over-smoothing and overfitting, making the model inefficient and computationally expensive. LightGCN[10] addresses these issues by simplifying the

architecture—removing the non-linear activation functions and transformation parameters typically used in GNN models. This reduction in complexity not only improves computational efficiency but also enhances the model's ability to focus on essential collaborative signals. However, such aggregation-based modeling paradigm does not explicitly account for the sequential or semantic continuity of nodes along interaction paths.

**Self-Supervised Learning Techniques in Recommender Systems.** In recent years, Self-supervised learning has become an important technique in enhancing recommendation models, especially when dealing with sparse or noisy data. In contrastive self-supervised learning, the goal is to pull positive samples closer together while pushing negative samples further apart, thus improving the robustness of learned representations. SimGCL[33] proposes a contrastive learning framework for graph-based recommendation models, where the authors introduce augmented views of users and items to enhance the model's ability to distinguish between relevant and irrelevant interactions. By incorporating contrastive loss, SimGCL not only improves the robustness of the learned embeddings but also helps address the issue of negative sampling, which is often problematic in collaborative filtering-based systems. Furthermore, NCL[14] enhances this approach by clustering users and items into meaningful groups, allowing the model to leverage these clusters as positive samples. This clustering technique, combined with contrastive loss, significantly improves the quality of the learned representations, particularly in sparse scenarios. IHGCL[19] leverages meta-paths in heterogeneous graphs to extract user intents and enhances recommendation via intent–intent and intent–interaction contrastive learning. However, existing work does not fully consider the relationships between target nodes and nodes on the interaction path in a recommendation scenario. In this paper, we use contrastive learning to interpretively model these potential node relationships.

## 3 The PECL framework

In this section, we introduce the PECL framework. The PECL framework is composed by four parts: graph collaborative filtration backbone LightGCN, path node-aware contrastive learning method, path sampling network and the path-aware contrastive learning method. The overall framework is shown in Fig.1.

### 3.1 Graph Collaborative Filtering Backbone

In PECL, we use LightGCN as the backbone model. The LightGCN model focuses on effectively capturing high-order collaborative signals in a user-item interaction graph by leveraging a simplified GCN structure. Given a bipartite graph $\mathcal{G} = (\mathcal{V}, \mathcal{E})$, the node set $\mathcal{V} = \mathcal{U} \cup \mathcal{I}$ includes both users and items, and the edge set $\mathcal{E} = \{(u, i) | \mathcal{R}_{ui} = 1\}$ captures observed interactions $\mathcal{R}_{ui}$. that connects users and items. LightGCN updates node embeddings over $k$ layers, where each layer aggregates information from increasingly distant neighbors. The embedding at the $k$-th layer, $\mathbf{H^k}$ represents knowledge collected from $k$-hop neighbors. To counteract the over-smoothing commonly observed in deep GCNs, LightGCN constructs the final node representations by combining embeddings from all layers, including the input embeddings:

$$\begin{aligned} \mathbf{H} &= \alpha_0\mathbf{H^{(0)}} + \alpha_1\mathbf{H^{(1)}} + \alpha_2\mathbf{H^{(2)}} + \ldots + \alpha_K\mathbf{H^{(K)}} \\ &= \alpha_0\mathbf{H^{(0)}} + \alpha_1\tilde{\mathbf{A}}\mathbf{H^{(0)}} + \alpha_2\tilde{\mathbf{A}}^2\mathbf{H^{(0)}} + \ldots + \alpha_K\tilde{\mathbf{A}}^K\mathbf{H^{(0)}}, \end{aligned} \tag{1}$$

$$\tilde{\mathbf{A}} = \mathbf{D}^{-\frac{1}{2}}\mathbf{A}\mathbf{D}^{-\frac{1}{2}}, \tag{2}$$

where $\tilde{\mathbf{A}}$ is the symmetrically normalized matrix and $\mathbf{D}$ is a $(|\mathcal{U}| + |\mathcal{I}|) \times (|\mathcal{U}| + |\mathcal{I}|)$ diagonal matrix. The 0-th layer embedding matrix $\mathbf{H^{(0)}} \in \mathbb{R}^{(|\mathcal{U}|+|\mathcal{I}|)\times E}$, where $E$ is the embedding size.

These final embeddings are used to represent users and items—specifically, user $u$ and item $i$ are represented by $h_u$ and $h_i$, respectively. The predicted interaction score between user $u$ and item $i$ is then computed using the inner product operation:

$$\hat{r}_{ui} = h_u h_i^{\mathrm{T}} \tag{3}$$

The model is trained using a ranking loss function, encouraging observed interactions (positive samples from $\mathcal{R}^+$) to have higher scores than unobserved ones (negative samples from $\mathcal{R}^-$), with a sigmoid function $\sigma(x) = 1/(1 + e^{-x})$ used to map predictions to probabilities:

$$\mathcal{L}_{\text{BPR}} = -\sum_{u\in\mathcal{U}}\sum_{i\in\mathcal{R}_u^+, j\in\mathcal{R}_u^-} \log(\sigma(\hat{r}_{ui} - \hat{r}_{uj})) \tag{4}$$

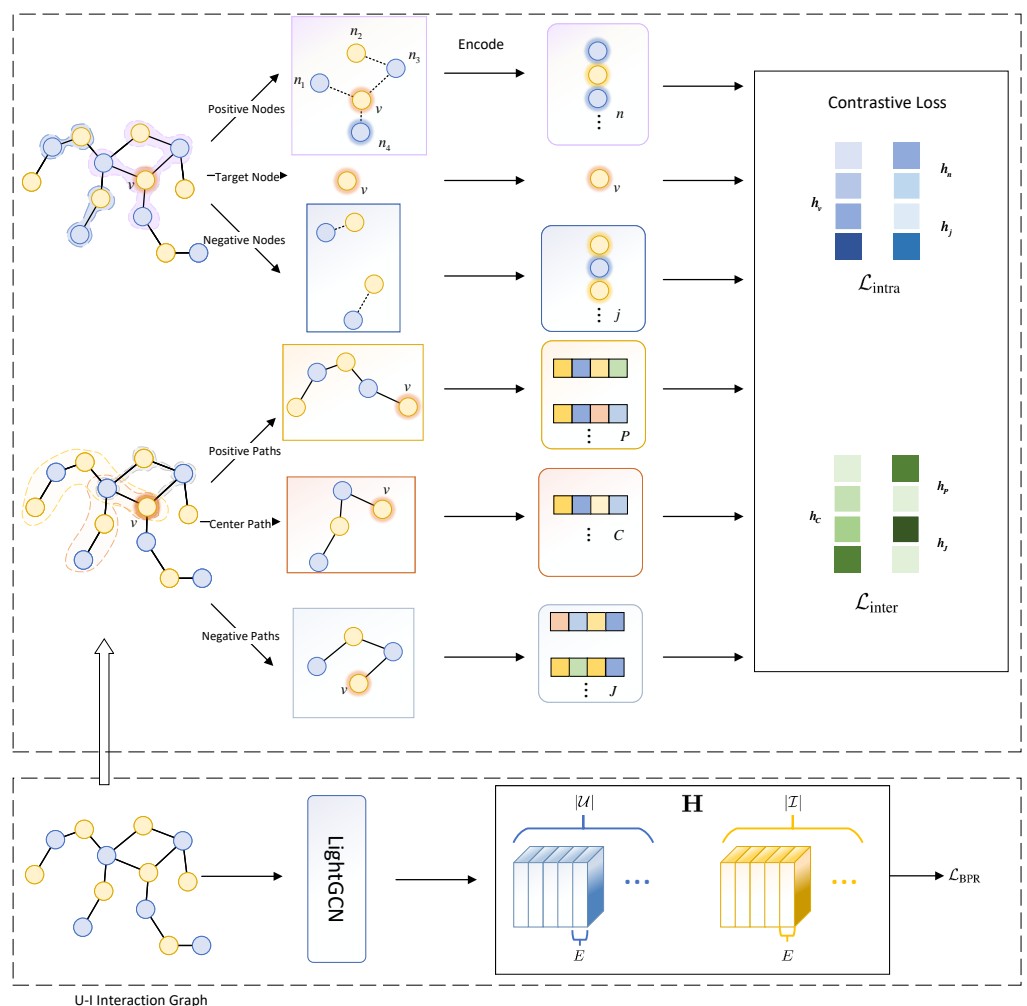

Figure 1: Method framework.

## 3.2 Path Node Aware Contrastive Learning

Existing graph augmentation contrastive learning methods mainly augment graph collaborative filtering by utilizing similar or structural neighbors[14, 22]. These methods ignore the internal nodes of the interaction paths, however, in fact, the nodes inside the interaction paths also have influence on the target nodes. In order to take full advantage of contrastive learning, we argue that when constructing the contrast loss of a target node, it is also important to consider the representatives of the nodes inside its interaction path as positive samples. Computing all nodes on paths may make the computation too large, so we use the random walk with restart for path sampling based on length classification proposed by Xiong et al.[30].

$$
p = \begin{cases} p_r, & \text{restart;} \\ \frac{1-p_r}{|\mathcal{N}_{neighbor}|}, & \text{random selection of neighbors,} \end{cases} \tag{5}
$$

where $\mathcal{N}_{neighbor}$ denotes the set of neighbors and the probability $p$ of the next move in a random walk is governed by two distinct scenarios. We obtain our set of center paths $\mathcal{P}_v$ of the target node $v$ utilizing the above sampling approach. In order to find the positive nodes $n$ of the target node $v$,we define a positive node selection method:

$$
\mathcal{V}_v^+ = \{n \,|\, \alpha \le |\{P \in \mathcal{P}_v \,|\, n \in P\}|\}, \tag{6}
$$

where $P$ denotes the path and $\alpha$ is a hyperparameter that determines the minimum number of positive nodes in sample paths. Then, other nodes are treated as negative samples. During the contrast loss construction process, specifically, we learn the embeddings of the target nodes themselves and the sampled intra-path nodes by contrast. Building on the principles of InfoNCE[16], we introduce an intra-path contrastive learning framework aimed at encouraging closer alignment between related representations by reducing their mutual divergence:

$$\mathcal{L}_{\text{intra}} = -\sum_{v \in \mathcal{V}} \log \sum_{n \in \mathcal{V}_v^+} \frac{\exp(h_v(h_n)^{\text{T}}/\tau)}{\sum_{j \in \mathcal{V}} \exp(h_v(h_j)^{\text{T}}/\tau)}, \tag{7}$$

where $h$ denotes the embedding vector of the node and $j$ denotes the negative node.

### 3.3 Preparation for Path Aware Contrastive Learning

This section will introduce how to find sample paths of the path-aware contrastive learning. Our ultimate target is to bring the target node on the center path closer to the nodes on the positive path through the path. Random walk sampling with the target node as the starting point is an option. However, since we define our own way of obtaining the center path, the path obtained by random walk sampling is not guaranteed to have a high similarity with our center path, since it can only determine that it has the same starting point. The meta-path sampling approach is not available for the recommendation task we study because we only have a simple user-item binary graph. To our knowledge, few studies have performed contrastive learning between paths, while our study has center paths that we define, so we need a corresponding positive path sampling method which can ensure that the computation is not too large and the similarity is also good. Meanwhile, negative sampling is performed in other paths.

To better capture the time dynamics and structural patterns in user-item interaction graphs, we propose a two-stage path sampling method termed Target-guided Random Walk. This strategy integrates deterministic time traversal with stochastic exploration, enabling sampled paths to reflect both the structured information flow and the inherent uncertainty of user behavior.

Formally, each edge $(u, i) \in \mathcal{E}$ is associated with a timestamp $t(u, i) \in \mathcal{T}$. For the set of center paths $\mathcal{P}_v$ of the target node $v$, we define a center path $C = (v_1, v_2, \ldots, v_n), C \in \mathcal{P}_v$, constructed by following the time interaction sequence such that:

$$t(v_i, v_{i+1}) \le t(v_{i+1}, v_{i+2}), \quad \forall i = 1, \ldots, n-2 \tag{8}$$

This path reflects a plausible trajectory of user or item interaction based on actual historical data.

We divide the sampling process into two stages to obtain path $C$'s positive path set $\mathcal{P}_C^+$:

Stage 1 (Target-guided traversal): We deterministically follow one center path $C \in \mathcal{P}_v$ from the start node $v_1$ to the intermediate node $v_\beta$, where $\beta$ is a hyperparameter controlling the number of steps before random walking. The partial path is denoted as:

$$P^{(1)} = (v_1, v_2, \ldots, v_\beta) \tag{9}$$

Stage 2 (Conditional random walk): Starting from node $v_\beta$, we conduct a time-aware conditional random walk to generate the remaining $n - \beta$ nodes. At each step $j$, the next node $v'_{j+1}$ is sampled from the neighborhood $\mathcal{N}(v'_j)$ based on a temporally-biased transition probability $\mathcal{TP}(v'_j)$:

$$\mathcal{TP}(v'_j) = \text{Softmax}_{v \in \mathcal{N}(v'_j)} \left( \frac{1}{\Delta t(v'_j, v)} \cdot w(v'_j, v) \right), \tag{10}$$

where $\Delta t(v'_j, v) = t(v'_j, v) - t_{\text{last}}$ captures the relative time gap, and $w(v'_j, v)$ denotes a tunable edge weight that can incorporate interaction frequency or node similarity. The second-stage path $P^{(2)}$ is defined as a sequence of nodes generated via a time-aware random walk process:

$$P^{(2)} = (v'_{\beta+1}, \ldots, v'_n), \quad \text{where} \quad v'_{j+1} \sim \mathcal{TP}(v'_j) \quad \text{for } j = \beta, \ldots, n-1 \tag{11}$$

The final sampled path is defined as:

$$P = P^{(1)} \cup P^{(2)} = (v_1, \ldots, v_\beta, v'_{\beta+1}, \ldots, v'_n) \tag{12}$$

This hybrid sampling mechanism ensures that the sampled paths maintain time consistency with real-world interaction sequences while introducing stochasticity to enhance path diversity and coverage.

## 3.4 Path Aware Contrastive Learning

The inter-path loss introduced in this section leverages representative paths extending from the target node as positive counterparts of the center path, enabling the model to align semantically similar paths and mitigate the ambiguity of single-node semantics.

To perform inter-path contrastive learning, we need to encode the paths. Since we are sampling the paths based on time series, we consider incorporating time information in the encoding of the paths. We use Temporal Context Encoding[11] to get the encoding of the timestamps $e$. We show specific coding methods in A.5. Since the timestamp in the recommender system is an attribute of the user-item relationship, i.e., it is a time code corresponding to each edge on the graph. So it is necessary to choose a suitable way to fuse node and edge information. In this paper, the Hermitian inner product is chosen as the interaction between node embedding and time embedding. The rotation on the complex plane maintains the invariance of the relationship between the node embedding and the time embedding, so that the distance between the two in the complex plane does not change due to the rotation. Normalizing the time embedding and then doing the Hermitian inner product operation with the node embedding, i.e., rotating the node embedding by a certain angle in the complex plane. For node embeddings and time embeddings, first treat the first half of the dimension of the embedding vector as the real part $x^{real}, e^{real}$ and the second half of the dimension as the imaginary part $x^{img}, e^{img}$. Perform the Hermitian inner product with the edges from the farthest node of the path towards the center node in the following order:

$$
\begin{aligned}
w_0 &= x_0', \\
w_{i-1}'^{real} &= w_{i-1}^{real} \odot e_i^{real} + w_{i-1}^{img} \odot e_i^{img}, \\
w_{i-1}'^{img} &= -w_{i-1}^{real} \odot e_i^{img} + w_{i-1}^{img} \odot e_i^{real}, \\
w_{i-1}' &= \left( w_{i-1}'^{real} \parallel w_{i-1}'^{img} \right) \\
w_i &= x_i' + w_{i-1}', \\
h_P &= \frac{w_n}{n},
\end{aligned}
\tag{13}
$$

where $\odot$ denotes the elemental product of vectors, $w_i$ is the intermediate computation, $i$ denotes the order of nodes on the path, $i \in [0, L]$, and $L$ is the path length.

Similar to Eq.7, the contrast loss between paths is defined as follows:

$$
\mathcal{L}_{\text{inter}} = -\sum_{v \in \mathcal{V}} \sum_{C \in \mathcal{P}_v} \log \sum_{P \in \mathcal{P}_C^+} \frac{\exp(h_C(h_P)^{\text{T}}/\tau)}{\sum_{J \in \mathcal{P}_v} \exp(h_C(h_J)^{\text{T}}/\tau)}
\tag{14}
$$

## 3.5 Gradient Analysis and Comparative Study of Contrastive Losses

Contrastive learning enhances representation learning by encouraging proximity between positive samples and separation from negatives. In this section, we provide a detailed gradient derivation for the intra-path and inter-path contrastive losses and analyze their roles in optimization.

**Gradient of Intra-path Contrastive Loss.** For the gradient of the intra-path loss with respect to the target node embedding, we differentiate the Eq.7:

$$
\frac{\partial \mathcal{L}_{\text{intra}}}{\partial h_v} = -\sum_{n \in \mathcal{V}_v^+} \frac{\exp(h_v h_n^{\text{T}}/\tau)}{Z_{\text{intra}}} \frac{h_n}{\tau} + \sum_{j \in \mathcal{V}} \frac{\exp(h_v h_n^{\text{T}}/\tau)}{Z_{\text{intra}}} \frac{h_j}{\tau}
\tag{15}
$$

where $Z_{\text{intra}}$ is the partition function given by:

$$
Z_{\text{intra}} = \sum_{n \in \mathcal{V}_v^+} \exp(h_v h_n^{\text{T}}/\tau) + \sum_{j \in \mathcal{V}} \exp(h_v h_n^{\text{T}}/\tau)
\tag{16}
$$

The gradient analysis reveals that the intra-path loss encourages positive nodes closer to the target node while pushing negative nodes away. The strength of this attraction-repulsion mechanism is modulated by the temperature parameter $\tau$.

**Gradient of Inter-path Contrastive Loss.**    Recalling the inter-path loss from Eq.14, we compute the gradient w.r.t. with respect to the center path embedding $h_C$:

$$\frac{\partial \mathcal{L}_{\text{inter}}}{\partial h_C} = -\sum_{P \in \mathcal{P}_C^+} \frac{\exp(h_C h_P^{\text{T}}/\tau)}{Z_{\text{inter}}} \frac{h_P}{\tau} + \sum_{J \in \mathcal{P}} \frac{\exp(h_C h_J^{\text{T}}/\tau)}{Z_{\text{inter}}} \frac{h_J}{\tau} \tag{17}$$

where $Z_{\text{inter}}$ is the partition function defined as:

$$Z_{\text{inter}} = \sum_{P \in \mathcal{P}_C^+} \exp(h_C h_P^{\text{T}}/\tau) + \sum_{J \in \mathcal{P}} \exp(h_C h_J^{\text{T}}/\tau) \tag{18}$$

The gradient analysis indicates that the inter-path loss minimizes the discrepancy between the center path and positive paths while distinguishing them from negative paths.

From the gradient perspective, the intra-path loss directly updates node embeddings through their pairwise similarities weighted by influence functions, resulting in localized embedding refinement. In contrast, the inter-path loss updates path-level representations that indirectly influence the node embeddings through path encoding networks, promoting global consistency across paths. Combining these two losses yields a complementary synergy: intra-path loss strengthens local neighborhood coherence within sampled paths, while inter-path loss encourages global structural alignment across diverse paths. This joint optimization facilitates learning robust and discriminative representations for recommendation tasks. We provide further analysis in A.6.

### 3.6   Model Training

In this section, we will present the overall loss of the model. As with most contrastive learning methods, our model loss is composed of the BPR loss for interactions between user items and the contrastive learning loss.

$$\mathcal{L} = \mathcal{L}_{\text{BPR}} + \lambda_1 \mathcal{L}_{\text{intra}} + \lambda_2 \mathcal{L}_{\text{inter}} + \lambda_3 \|\Theta\|_2 \tag{19}$$

where $[\lambda_1, \lambda_2, \lambda_3]$ denotes the regularized penalty coefficient and $\Theta$ corresponds to the parameters of the model. We give the values of lambda in A.7.

## 4   Experimental Results

### 4.1   Experimental Settings

Our experiments utilize three publicly available, real-world datasets—ML-1M, Ciao and Amazon—that offer rich information, including both user-item ratings and the corresponding timestamps of user interactions. Each dataset is divided into training and test sets by randomly selecting 80% of the rating entries for training purposes, while the leftover 20% is reserved to evaluate the model's performance during testing. To assess the effectiveness of our recommendation approach, we rely on two widely adopted evaluation metrics: Recall@K and NDCG@K. In our experiment, K is set to 10 and 20. To evaluate performance differences, we conduct a comparison between PECL and nine state-of-the-art recommendation approaches: SimGCL[33], SGL[26],SCCF[29], LightGCN[10], NGCF[25], SelfCF[34], SEPT[32], GAIPSRec[30], IHGCL[19].

### 4.2   Comparative Experiments

To validate the effectiveness and generalizability of the proposed PECL, we conducted a comprehensive performance comparison against several baseline models using three distinct datasets. We compute the average results 5 times for each dataset. The results of these experiments are presented in Table 1, and the key findings are summarized as follows: The experimental results demonstrate that PECL outperforms all baseline models in both Recall@K and NDCG@K across the evaluated datasets. In particular, PECL achieves significant gains over the best-performing baseline models in terms of NDCG@10 on the Ciao, with improvement rate of 4.52%. The improvement can be attributed to the following key factors: (1) PECL generates multiple complementary nodes and paths through its contrastive learning framework. These nodes and paths are not simply based on

| Datasets | Metrics | NGCF | SGL | LightGCN | SEPT | SelfCF | SimGCL | SCCF | GAIPSRec | IHGCL | PECL |
|---|---|---|---|---|---|---|---|---|---|---|---|
| ML-1M | NDCG@10 | 0.3279 | 0.3645 | 0.3542 | 0.3529 | 0.3629 | 0.3667 | 0.3364 | 0.4096 | 0.4148 | **0.4206** |
| | NDCG@20 | 0.3115 | 0.3981 | 0.3476 | 0.3462 | 0.3475 | 0.3523 | 0.3274 | 0.3985 | 0.4006 | **0.4024** |
| | Recall@10 | 0.1275 | 0.1429 | 0.1443 | 0.1357 | 0.1514 | 0.1542 | 0.1437 | 0.1582 | 0.1601 | **0.1676** |
| | Recall@20 | 0.2161 | 0.2510 | 0.2419 | 0.2218 | 0.2259 | 0.2373 | 0.2501 | 0.2514 | 0.2565 | **0.2605** |
| Ciao | NDCG@10 | 0.0298 | 0.0376 | 0.0354 | 0.0354 | 0.0371 | 0.0369 | 0.0297 | 0.0362 | 0.0364 | **0.0393** |
| | NDCG@20 | 0.0353 | 0.0447 | 0.0435 | 0.0368 | 0.0451 | 0.0471 | 0.0360 | 0.0473 | 0.0450 | **0.0486** |
| | Recall@10 | 0.0338 | 0.0483 | 0.0420 | 0.0433 | 0.0439 | 0.0476 | 0.0444 | 0.0481 | 0.0476 | **0.0496** |
| | Recall@20 | 0.0520 | 0.0713 | 0.0687 | 0.0678 | 0.0721 | 0.0721 | 0.0663 | 0.0728 | 0.0691 | **0.0745** |
| Amazon | NDCG@10 | 0.0785 | 0.1091 | 0.1016 | 0.1013 | 0.1054 | 0.1021 | 0.1086 | 0.1105 | 0.1132 | **0.1169** |
| | NDCG@20 | 0.0913 | 0.1256 | 0.1183 | 0.1156 | 0.1195 | 0.1243 | 0.1208 | 0.1240 | 0.1309 | **0.1354** |
| | Recall@10 | 0.0813 | 0.1135 | 0.1084 | 0.1034 | 0.1089 | 0.1145 | 0.1029 | 0.1128 | 0.1193 | **0.1241** |
| | Recall@20 | 0.1257 | 0.1501 | 0.1528 | 0.1486 | 0.1548 | 0.1691 | 0.1494 | 0.1664 | 0.1764 | **0.1820** |

Table 1: NDCG and Recall of PECL and baseline models.

random augmentations, but on the rich paths formed through both direct interactions and higher-order connections within the user-item graph. This mechanism ensures that PECL can effectively learn diverse representations that better capture the complex relationships between users and items. (2) Unlike traditional recommendation models that focus solely on direct user-item interactions, PECL introduces a novel contrastive learning strategy that incorporates path-based comparisons. By leveraging multiple interaction paths between users and items, PECL ensures that richer and more diverse representations of user preferences are captured. This path-based contrast helps the model discern nuanced patterns in the data, allowing for more accurate recommendations. (3) The combination of temporal information and collaborative filtering principles enables PECL to balance historical interaction patterns with the evolving preferences of users, resulting in improved predictive accuracy and user satisfaction.

When analyzing the performance of all baseline models, it is clear that self-supervised learning (SSL)-enhanced methods consistently outperform traditional recommendation approaches. This trend is especially noticeable in models such as SimGCL and SGL, which incorporate SSL techniques into the learning process. While these methods are useful, they fail to fully capture the complexity of user-item interactions and their higher-order relationships. On the other hand, PECL's path-based contrastive learning directly addresses this issue by comparing different paths within the user-item interaction graph. This allows PECL to learn richer and more meaningful representations.

## 4.3 Ablation Study

To verify the effectiveness of the core components in PECL, we conducted ablation studies by designing the following model variants: PECL w/o Inter-Path Contrast(w/o inter): This variant removes the inter-path contrastive learning mechanism. PECL w/o Intra-Path Contrast(w/o intra): This version excludes the intra-path contrast mechanism. PECL w/o Path Sampling(w/o PS): In this variant, we replace our proposed path sampling method with a standard random walk sampling technique. PECL (Full): The complete model incorporating all three components, serving as the baseline for comparison. The performance of these variants is summarized in Fig.2 in terms of Recall@10, NDCG@10 on the ML-1M and Ciao.

When the inter-path contrast mechanism is removed, the model experiences substantial performance degradation, with a decline of 27.4% in Recall@10 on ML-1M. This demonstrates that contrasting multiple paths effectively enhances the model's capability to capture diverse interaction patterns across different user-item paths, leading to more comprehensive and robust representations. The exclusion

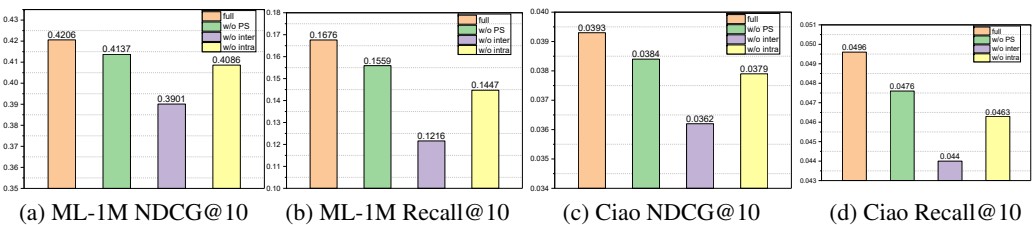

(a) ML-1M NDCG@10    (b) ML-1M Recall@10    (c) Ciao NDCG@10    (d) Ciao Recall@10

Figure 2: PECL and its variants.

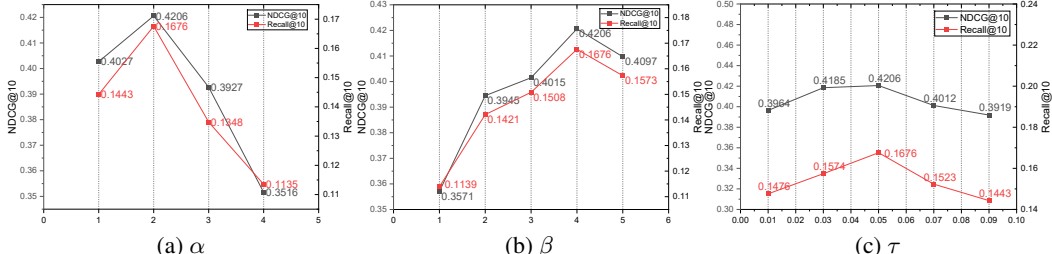

Figure 3: Effect of hyperparameters on ML-1M.

of intra-path contrast also results in noticeable performance drops, particularly in both metrics, where the model shows a 7.3% reduction in NDCG@10 and a 13.7% reduction on ML-1M. This indicates that contrasting within a single path helps the model refine its representations by reinforcing the consistency of the learned embeddings within each path, thus preventing noisy or trivial patterns from dominating. Replacing our path sampling method with random walk sampling (PECL w/o PS) leads to declines in both Recall and NDCG scores, with a 11.3% drop in Recall@10 on ML-1M. This demonstrates that our proposed path sampling strategy is more effective in selecting informative and diverse paths that contribute to more discriminative and expressive user-item representations. PECL w/o PS's decline is least because it does not ablate any contrast loss. The full model (PECL) consistently outperforms all ablation variants, indicating that each proposed component contributes to the overall performance improvement. Notably, the inter-path contrast mechanism exhibits the most significant impact, highlighting its crucial role in capturing diverse and informative paths.

## 4.4 Hyperparameter Analysis

To comprehensively evaluate the impact of hyperparameters on PECL performance, we performed a series of controlled experiments by varying key hyperparameters. Specifically, we focus on three crucial hyperparameters: $\alpha$ (Eq.6), $\beta$ (Eq.9), and $\tau$ (Eqs.7, 14). We conducted these experiments on the ML-1M with NDCG@10 and Recall@10. Experiments on Ciao are shown in A.8.5.

We vary $\alpha$ in $\{1, 2, 3, 4\}$ to assess how the number of positive nodes influences the model's ability to capture various interaction patterns. As depicted in Fig.3a, increasing the $\alpha$ from 1 to 2 improves the performance, with an 4.4% gain in NDCG@10 on ML-1M. However, further increasing $\alpha$ to 4 leads to performance decreases. When the value of alpha is too small, the model tends to capture too many positive nodes, leading to overfitting. Conversely, when alpha is too large, the model may fail to capture sufficient path nodes, resulting in potential information loss. Therefore, setting alpha to 2 can effectively balance the model's generalization ability and the capture of path nodes, thereby enhancing the overall performance.

We examine $\beta$ in $\{1, 2, 3, 4, 5\}$ to verify the effect of this parameter on our path sampling method. As shown in Fig.3b, increasing the number of positive nodes generally improves performance, with the optimal performance observed at $\beta = 4$. further increase $\beta$ beyond 4 leads to performance degradation. This is likely because excessive positive nodes introduce redundant, diluting the effectiveness of contrastive learning. It is worth noting that the performance improvement when changing from odd to even is higher when alpha is less than 5. This is due to the fact that in the user-item bipartite graph, paths grow even lengths with the addition of nodes of the same type, which allows for the learning of more efficient path node information.

We vary $\tau$ in $\{0.01, 0.03, 0.05, 0.07, 0.09\}$ to investigate its effect on representation discrimination. As depicted in Fig.3c, the impact of $\tau$ is evident in both metrics. When $\tau$ is set to 0.05, the model achieves the highest performance, indicating that a moderate temperature provides a balanced contrastive distribution. Setting $\tau$ too low results in overly sharp distributions, making it difficult for the model to differentiate similar paths. Conversely, a higher $\tau$ produces excessively smooth distributions, weakening the contrastive effect.

# 5 Conclusion

In this work, we proposed a novel path-enhanced contrastive learning framework for recommendation, which leverages the structural information embedded in multiple interaction paths to enhance representation learning. Unlike conventional contrastive learning methods that primarily focus on node-level or edge-level interactions, PECL explores both inter-path and intra-path relationships, effectively capturing richer contextual dependencies among users and items. Additionally, our tailored path sampling strategy mitigates the risk of redundant or noisy paths, enabling more informative contrastive learning signals. Extensive experiments conducted on three real world datasets, ML-1M, Ciao and Amazon, demonstrate the effectiveness of the proposed framework, highlighting the importance of path-level contrastive learning and the proposed path sampling strategy. In summary, this study underscores the potential of integrating path-level information into contrastive learning for recommendation tasks, offering a promising direction for further exploration. Future work may extend this framework by incorporating adaptive path sampling techniques or integrating additional auxiliary information to further improve recommendation accuracy and robustness.

## Acknowledgments

This work has been supported by the Fundamental Research Funds for the Central Universities under Grant 2024YJS203 and the National Natural Science Foundation of China under Grant 62472024.

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

# A Technical Appendices and Supplementary Material

## A.1 Notation and Description

| Symbol | Description |
|---|---|
| $\mathcal{U}$ | Set of users |
| $\mathcal{I}$ | Set of items |
| $\mathcal{R}$ | User-item interaction matrix |
| $\mathcal{G}$ | Bipartite graph representing interactions |
| $\mathcal{V}$ | Set of nodes (users and items) |
| $\mathcal{E}$ | Set of edges (interactions) |
| $\mathbf{H}^k$ | Node embeddings at layer $k$ |
| $\tilde{\mathbf{A}}$ | Symmetrically normalized adjacency matrix |
| $p_r$ | Restart probability in random walk |
| $\mathcal{N}_{neighbor}$ | Set of neighbors |
| $\tau$ | Temperature parameter in contrastive loss |
| $\mathcal{P}_v$ | Path set for node $v$ |
| $h_u, h_i$ | Embeddings of user $u$ and item $i$ |
| $\mathcal{L}_{\text{BPR}}$ | BPR loss function |
| $\mathcal{L}_{\text{intra}}$ | Intra-path contrastive loss |
| $\mathcal{L}_{\text{inter}}$ | Inter-path contrastive loss |

Table 2: Symbols and descriptions used in the PECL model

## A.2 Preliminary

CF is a foundational technique in recommender systems, designed to identify and suggest items that users are likely to engage with, inferred from implicit feedback such as clicks, purchases, or other forms of interaction. Consider a set of users $\mathcal{U} = \{u\}$ and a set of items $\mathcal{I} = \{i\}$. The user-item interaction data is captured in a binary matrix $\mathbf{R} \in \{0, 1\}^{|\mathcal{U}| \times |\mathcal{I}|}$, where an entry $\mathcal{R}_{ui} = 1$ indicates that user $u$ has interacted with item $i$, and $\mathcal{R}_{ui} = 0$ otherwise. Using this matrix $\mathbf{R}$, recommendation models aim to infer unobserved interactions and predict user preferences.

In recent approaches, GNNs have been leveraged to enhance collaborative filtering by modeling the user-item interactions as a bipartite graph $\mathcal{G} = (\mathcal{V}, \mathcal{E})$, where the node set $\mathcal{V} = \mathcal{U} \cup \mathcal{I}$ includes both users and items, and the edge set $\mathcal{E} = \{(u, i) | \mathcal{R}_{ui} = 1\}$ captures observed interactions. GNN-based methods learn expressive user and item embeddings by recursively aggregating information from neighboring nodes in the graph. Typically, this process involves two major phases: information propagation to propagate neighborhood signals, and update operations to generate refined representations:

$$
\begin{aligned}
h_u^{(k)} &= AGGREGATE^{(k)} \left( \left\{ h_v^{(k-1)} | v \in \mathcal{N}(u) \right\} \right), \\
h_u &= UPDATE^{(k)}(h_u^{(0)}, h_u^{(1)}, \dots, h_u^{(k)}),
\end{aligned}
\tag{20}
$$

where $h_u^{(k)}$ is the updated representation of node $u$ at layer $k$ and $\mathcal{N}(u)$ denotes the neighboring nodes of node $u$.

### A.3 Pesudo Code

---

**Algorithm 1:** PECL Framework Training Algorithm

---

**Input:** User-item interaction graph $\mathcal{G} = (\mathcal{U} \cup \mathcal{I}, \mathcal{E})$
Embedding dimension $E$, number of LightGCN layers $K$
Number of sampled paths per node $S$, path length $L$
Two-stage path sampling parameter $\beta$
Number of positive/negative samples $N_p, N_n$
Learning rate $\eta$, total training epochs $T$
**Output:** Learned embeddings $\mathbf{H}^*$ for users and items

---

1   Initialize embeddings $\mathbf{H}^{(0)}$ for all nodes randomly;

2   Precompute normalized adjacency matrix $\tilde{\mathbf{A}}$;

3   **for** $epoch = 1$ **to** $T$ **do**

     `// LightGCN Embedding Propagation`

4      $\mathbf{H}^{(0)} \leftarrow$ initial embeddings;

5      **for** $k = 1$ **to** $K$ **do**

6          $\mathbf{H}^{(k)} \leftarrow \tilde{\mathbf{A}} \times \mathbf{H}^{(k-1)}$;

7      Compute final embeddings:

$$\mathbf{H}^* = \sum_{k=0}^{K} \alpha_k \mathbf{H}^{(k)}$$

     where $\alpha_k$ are layer weights;

     `// Path Node Aware Contrastive Learning`

8      **foreach** *node* $v \in \mathcal{U} \cup \mathcal{I}$ **do**

9          $\mathcal{P}_v \leftarrow$ empty set to store sampled paths;

10         **for** $s = 1$ **to** $S$ **do**

11             $p \leftarrow$ Sample path of length $L$ from node $v$ via random walk with restart (temporal constraints);

12             Add $p$ to $\mathcal{P}_v$;

        `// Construct positive node set` $\mathcal{V}_v^+$

13         $\mathcal{V}_v^+ \leftarrow \{\}$;

14         **foreach** *path* $p$ *in* $\mathcal{P}_v$ **do**

15             For positive node $u$ in $p$, add $u$ to $\mathcal{V}_v^+$;

16         Sample $N_n$ negative nodes;

17         Compute intra-path contrastive loss $\mathcal{L}_{\text{intra}}$ using Eq.(7) on embeddings $\mathbf{H}^*$ for node $v$, $\mathcal{V}_v^+$, and negatives;

     `// Path Aware Contrastive Learning`

18      **foreach** *node* $v \in \mathcal{U} \cup \mathcal{I}$ **do**

19          $\mathcal{Q}_v \leftarrow$ empty set for sampled paths;

20         **for** $s = 1$ **to** $S$ **do**

21             $p \leftarrow$ Two-stage path sampling of length $L$ from node $v$:

                 • Deterministic walk $\beta$ steps along center path

                 • Conditional random walk for remaining $L - \beta$ steps;

            Add $p$ to $\mathcal{Q}_v$;

22         **foreach** $p_i$ *in* $\mathcal{Q}_v$ **do**

23             Sample negative path $p_j$;

24             Encode path into vector embeddings using temporal context encoding and Hermitian inner product;

25             Compute inter-path contrastive loss $\mathcal{L}_{\text{inter}}$ on pairs;

     `// Update embeddings`

26      Update $\mathbf{H}^{(0)}$ using gradients from $\mathcal{L}_{\text{intra}}$ and $\mathcal{L}_{\text{inter}}$ via optimizer with learning rate $\eta$;

27   **return** $\mathbf{H}^*$;

---

### A.4 Complexity Analysis

In this section, we analyze the computational complexity of the PECL framework, focusing on its main components: the LightGCN backbone, path node-aware contrastive learning, and the path-aware contrastive learning with path sampling.

#### A.4.1 Complexity of LightGCN Backbone

The LightGCN backbone operates on a bipartite graph $\mathcal{G} = (\mathcal{U} \cup \mathcal{I}, \mathcal{E})$ with $|\mathcal{U}|$ users, $|\mathcal{I}|$ items, and $|\mathcal{E}|$ edges representing observed interactions.

At each layer $k$ of LightGCN, the embedding update is a sparse matrix multiplication of the normalized adjacency matrix $\tilde{\mathbf{A}}$ with the embedding matrix $\mathbf{H}^{(k-1)}$. Since $\tilde{\mathbf{A}}$ has $2|\mathcal{E}|$ non-zero entries (due to the bipartite graph symmetry), the cost per layer is approximately:

$$O(|\mathcal{E}| \times E), \tag{21}$$

where $E$ is the embedding dimension.

For $K$ layers, the total complexity for embedding propagation is:

$$O(K \times |\mathcal{E}| \times E). \tag{22}$$

Finally, the weighted sum aggregation of embeddings from all layers has a cost of:

$$O((|\mathcal{U}| + |\mathcal{I}|) \times K \times E), \tag{23}$$

which is typically negligible compared to the sparse matrix multiplications.

#### A.4.2 Complexity of Path Node Aware Contrastive Learning

The path node-aware contrastive learning module involves:

- **Random walk with restart path sampling:** For each target node $v$, paths of length $L$ are sampled by random walks guided by temporal constraints. Assuming $S$ paths per node, the sampling complexity per node is roughly:

$$O(S \times L). \tag{24}$$

  Since the random walk step involves selecting neighbors, and the neighbor size is generally small compared to $|\mathcal{U}|$ or $|\mathcal{I}|$, this step is efficient.

- **Positive node selection and influence computation:** For each node $v$, the positive node set $\mathcal{V}_v^+$ is constructed by checking path membership. This involves checking $S$ paths of length $L$, thus:

$$O(S \times L). \tag{25}$$

- **Contrastive loss computation:** The intra-path contrastive loss in Eq.(7) involves computing dot products between the target node and positive nodes, and between the target node and all negative nodes. Let $N_p = |\mathcal{V}_v^+|$ be the number of positive nodes and $N_n$ the number of negative samples per node. The cost per node is:

$$O((N_p + N_n) \times E). \tag{26}$$

  Typically, $N_p$ and $N_n$ are kept small by sampling to maintain efficiency.

Overall, the complexity of this module is:

$$O(|\mathcal{U}| + |\mathcal{I}|) \times \big( S \times L + (N_p + N_n) \times E \big). \tag{27}$$

#### A.4.3 Complexity of Path Aware Contrastive Learning with Path Sampling

The path-aware contrastive learning involves:

- **Two-stage path sampling:** - The first stage deterministically traverses $\beta$ nodes along the center path, costing:

$$O(\beta). \tag{28}$$

- The second stage performs conditional random walks of length $L - \beta$, each step involving a softmax over the neighbors. Assuming average degree $d$, the complexity per step is:

$$O(d), \tag{29}$$

thus for the random walk:

$$O((L - \beta) \times d). \tag{30}$$

The total sampling complexity per path is:

$$O(\beta + (L - \beta) \times d). \tag{31}$$

- **Path encoding and contrastive loss:** Encoding a path of length $L$ using temporal context encoding and Hermitian inner products involves:

$$O(L \times E) \tag{32}$$

operations per path.

The inter-path contrastive loss computes similarity between pairs of paths or between a center path and positive paths. If $M$ positive paths are sampled per center path, the cost per target path is:

$$O(M \times L \times E). \tag{33}$$

Considering all target nodes and their sampled paths, the total complexity is approximately:

$$O((|\mathcal{U}| + |\mathcal{I}|) \times (S \times (\beta + (L - \beta)d) + M \times L \times E)). \tag{34}$$

### A.4.4 Summary

The overall computational complexity of PECL is dominated by the LightGCN embedding propagation and the path-aware contrastive learning components. Given typical sparse user-item graphs where $|\mathcal{E}| \gg |\mathcal{U}| + |\mathcal{I}|$, and carefully controlled sampling parameters $S$, $L$, $\beta$, $M$, $N_p$, and $N_n$, the model remains scalable and efficient for large-scale recommendation tasks.

### A.5 Encoding of Timestamps

The time embedding is computed as follows:

$$e\left[k\right] = \begin{cases} \sin\left(\frac{timestamp}{10000}\right)^{2k/d}, k\%2 = 0 \\ \cos\left(\frac{timestamp}{10000}\right)^{2(k-1)/d}, k\%2 \neq 0 \end{cases} \tag{35}$$

where $k$ is a certain dimension of temporal embedding $e$, $d$ denotes the dimension of temporal embedding, and timestamp denotes the timestamp. The model can learn certain temporal dependencies of information dissemination paths through the temporal context of $t$.

### A.6 Comparison of the Two Loss Functions

| Aspect | Intra-path Contrastive Loss | Inter-path Contrastive Loss |
|---|---|---|
| Granularity | Node-level | Path-level |
| Positives Definition | Nodes frequently appearing in paths | Paths with spatiotemporal consistency |
| Optimization | Pulls nodes closer within same path | Pulls similar paths together, separates others |
| Computational Cost | Lower (scales with node set) | Higher (depends on path sampling) |
| Expressiveness | Local semantic focus | Global structural discrimination |

Table 3: Comparison of the Two Loss Functions

The gradient expressions of both losses show a common structure: positive embeddings attract while negative embeddings repel. The interaction between these two contrastive losses provides a

multi-level optimization strategy: the intra-path loss fine-tunes local relations between nodes, while the inter-path loss preserves the global path consistency. The balance between these gradients is controlled by the temperature $\tau$, which affects the sharpness of the similarity distribution. Both intra-path and inter-path contrastive losses are designed to improve the quality of node representations by leveraging structural information in the user-item interaction graph, but they operate at different granularities:

- **Intra-Path Loss** focuses on the local structure within a single path by contrasting the target node against nodes inside its sampled paths. It captures fine-grained relationships and encourages tight clustering of nodes that co-occur frequently within interaction paths. This helps the model learn nuanced user-item correlations informed by intermediate nodes on the paths.

- **Inter-Path Loss**, on the other hand, operates at the path level by contrasting entire paths against each other. This loss mitigates the randomness and potential noise introduced by path sampling by encouraging the model to pull positive paths closer and push apart negative or unrelated paths. It effectively smooths the representation space by aligning semantically or temporally related paths.

### A.7 Gradient Normalization Strategy for Balancing Loss Terms

To ensure balanced optimization when combining the BPR loss with contrastive components, we adopt a gradient normalization strategy to determine the weighting coefficients $\lambda_1$ and $\lambda_2$ in the total loss Eq.19. This strategy ensures that the gradients of all loss components have similar magnitudes during backpropagation, thereby avoiding dominance by any single term. The weighting coefficients are computed as follows:

$$\lambda_1 = \frac{\|\nabla_\Theta \mathcal{L}_{\text{BPR}}\|_2}{\|\nabla_\Theta \mathcal{L}_{\text{intra}}\|_2}, \quad \lambda_2 = \frac{\|\nabla_\Theta \mathcal{L}_{\text{BPR}}\|_2}{\|\nabla_\Theta \mathcal{L}_{\text{inter}}\|_2} \tag{36}$$

Here, $\Theta$ denotes the full set of model parameters, and $\nabla_\Theta \mathcal{L}_i$ is the gradient of loss component $\mathcal{L}_i$ with respect to $\Theta$. Each gradient norm is computed as:

$$\|\nabla_\Theta \mathcal{L}_i\|_2 = \sqrt{\sum_{p \in \Theta} \left( \frac{\partial \mathcal{L}_i}{\partial p} \right)^2} \tag{37}$$

To reduce the variance caused by mini-batch noise, we optionally apply exponential moving average smoothing over multiple batches:

$$\bar{g}_i^{(t)} = \alpha \cdot g_i^{(t)} + (1 - \alpha) \cdot \bar{g}_i^{(t-1)} \tag{38}$$

where $\alpha \in [0.9, 0.99]$ is the decay rate, and $g_i^{(t)}$ denotes the raw gradient norm at batch $t$.

This normalization-based tuning of $\lambda_1$ and $\lambda_2$ leads to more stable convergence and improved model performance by balancing the optimization pressures from different loss components.

### A.8 Experimental

#### A.8.1 Device

All experiments we conducted are performed with Intel(R) Xeon(R) Bronze 3204 CPU @ 1.90GHz, Tesla A100, and 256GB memory, running the Ubuntu 20.04.4 LTS operating system.

#### A.8.2 Datases

We present information on the three real-world datasets we use in Table4.

| | ML-1M | Ciao | Amazon |
|---|---|---|---|
| Number of users | 6,040 | 101,998 | 6,170 |
| Number of items | 3,706 | 5,441 | 2,753 |
| Number of ratings | 1,000,209 | 268,975 | 195,791 |
| Sparrsity | 95.53% | 99.95% | 98.85% |

Table 4: Features of the evaluation datasets

### A.8.3 Baselines

**SimGCL[33]** introduces contrastive learning into graph collaborative filtering by generating perturbed graph views and maximizing agreement between them. It effectively enhances graph-based recommendation by mitigating oversmoothing issues.

**SGL[26]** employs contrastive learning to leverage both graph structure and node attributes, creating multiple views of the graph to learn robust user and item embeddings.

**SCCF[29]** unifies contrastive learning and collaborative filtering by jointly optimizing self-supervised signals and recommendation loss, enhancing robustness and generalization.

**LightGCN[10]** simplifies traditional GCNs by removing unnecessary components such as feature transformation and nonlinear activation, focusing purely on message passing to improve recommendation performance.

**NGCF[25]** integrates user-item interaction data into graph neural networks, allowing for richer and deeper embeddings that capture higher-order connectivity patterns.

**SelfCF[34]** adopts self-supervised contrastive learning to generate self-augmented views for collaborative filtering, aiming to mitigate data sparsity and improve recommendation accuracy.

**SEPT[32]** leverages contrastive learning to generate positive and negative samples for self-supervised learning, effectively improving collaborative filtering by aligning and contrasting multiple graph views.

**GAIPSRec[30]** introduces a graph attention network that captures high-order neighbor information through a heterogeneous graph sampling and path aggregation mechanism. It leverages attention and gating to integrate multi-hop dependencies and mitigate over-smoothing.

**IHGCL[19]** introduces an intent-guided heterogeneous graph contrastive learning framework. It leverages meta-path based intent representations with dual contrastive learning and employs a bottlenecked autoencoder to reduce noise, improving recommendation accuracy.

### A.8.4 Ablation Study

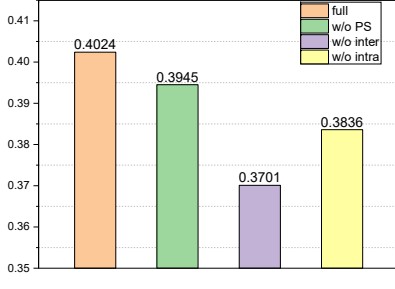

(a) ML-1M NDCG@20

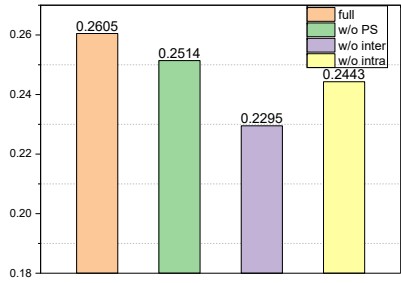

(b) ML-1M Recall@20

Figure 4: PECL and its variants on ML-1M.

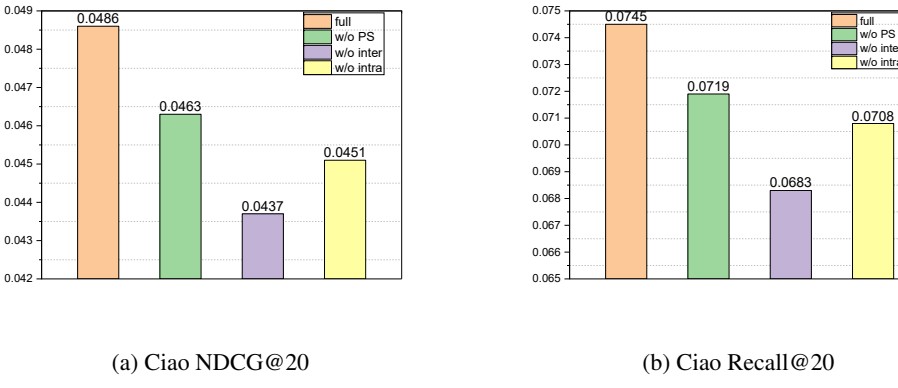

(a) Ciao NDCG@20                     (b) Ciao Recall@20

Figure 5: PECL and its variants on Ciao.

We supplement the top-20 results from the ablation experiments(Figs.4,5) with experimental conclusions consistent with the main text section.

### A.8.5  Hyperparameter Analysis

The experimental supplement for hyperparameters on the Ciao dataset is showed in Figs. 678.

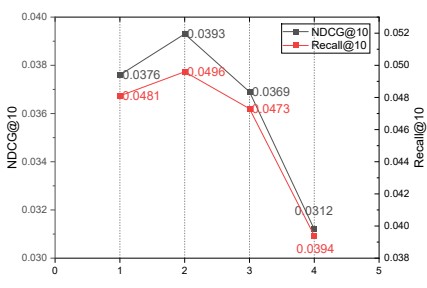

Figure 6: $\alpha$ on Ciao

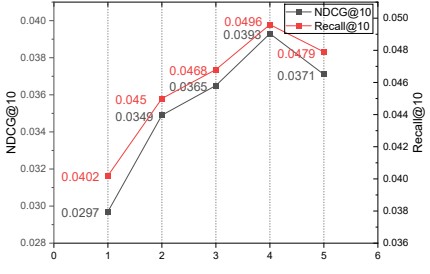

Figure 7: $\beta$ on Ciao

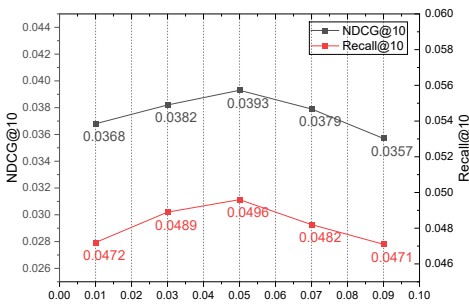

Figure 8: $\tau$ on Ciao

In Fig.6, we can see that the performance growth when alpha changes from 1 to 2 is not very much. However, if we fix alpha to 1, we can eliminate the positive node sampling work (Eq.6), which

reduces a certain computational complexity, and this problem can be analyzed specifically in a specific scenario.

### A.9 Ethic Impact Statements

Our paper uses real-world, publicly available datasets that do not raise ethical impact.

