# OpenReview forum: "Path-Enhanced Contrastive Learning for Recommendation"
_NeurIPS.cc/2025/Conference — NeurIPS 2025 spotlight_

### Official Review · Reviewer_8aC1 · 2025-06-19

**Clarity:** 2
**Significance:** 2
**Originality:** 2
**Rating:** 4
**Confidence:** 4

**Summary:**

This work proposes a path-enhanced contrastive learning for graph-based recommendation. Specifically, the authors consider graph-related augmentations based on both intra-path and inter-path correlations for better representation learning. In experiments, the authors have conducted evaluations on two datasets and achieved better results compared the baselines. They also verify the effectiveness of their two CL strategies in ablation study.

**Questions:**

Refer to Weaknesses.

**Ethical Concerns:**

["NO or VERY MINOR ethics concerns only"]

**Final Justification:**

I have read all reviews and the authors' rebuttal.

My original concerns mainly include (a) the insufficient related work, (b) the lack of recent strong GCL-related baselines, (c) the detailed baselines' settings and strange comparative results, (d) the lack of evaluation datasets (only two).

The author rebuttal has addressed most of my questions with additional experiments on two added baselines and one additional dataset. The discussions on the performance of models are also insightful. It is strongly suggested that these additional results and discussions should be included in the revision version. In conclusion, I will raise my voting to 4.

**Limitations:**

Yes

**Quality:**

2

**Strengths And Weaknesses:**

### Strengths:

1) The idea is sound and the writing is clear.
2) The authors have conducted ablation study and parameter analysis to better understanding the effects of different components and parameters in their model.

### Weaknesses:

1) The major weakness of this work is the lack of sufficient related work that also concentrate on graph contrastive learning in recommendation. There are lots of GCL related works for recommendation and other fields [1-5], some of which also consider path-related information to build positive/negative samples in CL. The authors should (i) appropriately cite and introduce these related efforts, giving the detailed pros and cons of their proposed inter-/intra- CL compared to existing methods, and (ii) compare with recent good-performing GCL methods (at least better than the classical backbone lightGCN on the evaluation datasets to ensure the persuasiveness of the conclusion).
2) Are the current baselines strong enough and were they properly trained? It is noticed that most of the compared baselines are proposed 2 years ago. The performance of SCCF (most recent baseline) is relatively poor (basically inferior to simGCL and even lightGCN), while simGCL and lightGCN are also comparable. The authors need to explain these unsatisfactory results of GCL baselines, and it is also recommended that the author release the codes and specific settings of essential baselines for reproducibility.
3) Whether the performance differences mainly derive from the amount of augmentation samples rather than the specific augmentation methods? The authors could show the number of augmentation samples of different methods, as well as their real training costs.
4) The current evaluations are mainly based on two datasets, and more datasets are required.
5) Whether PECL w/o intra-path and inter-path CL equals LightGCN? If so, it seems that both intra-path and inter-path CL strategies have nearly 4-6% improvement in NDCG@10 in ML-1M, while other baselines’ CL strategies have far less advantages. More discussions are needed.


### References:

[1] Xu L, Liu Y, Xu T, et al. Graph augmentation empowered contrastive learning for recommendation[J]. ACM Transactions on Information Systems, 2025, 43(2): 1-27.

[2] Huang J, Cao Q, Xie R, et al. Adversarial learning data augmentation for graph contrastive learning in recommendation[C]//International Conference on Database Systems for Advanced Applications. Cham: Springer Nature Switzerland, 2023: 373-388.

[3] Zhang S, Chen L, Wang C, et al. Temporal graph contrastive learning for sequential recommendation[C]//Proceedings of the AAAI Conference on Artificial Intelligence. 2024, 38(8): 9359-9367.

[4] Zhang W, Yang L, Song Z, et al. Mixed Supervised Graph Contrastive Learning for Recommendation[J]. arXiv preprint arXiv:2404.15954, 2024.

[5] Sang L, Wang Y, Zhang Y, et al. Intent-guided Heterogeneous Graph Contrastive Learning for Recommendation[J]. IEEE Transactions on Knowledge and Data Engineering, 2025.

---

> ### Author Rebuttal · Authors · 2025-07-31
>
> We thank reviewer for the constructive comments. We provide our feedbacks as follows.
>
> W1’s Response: We appreciate the opportunity to clarify the differences between our proposed Path-Enhanced Contrastive Learning (PECL) method and the related approaches you mentioned, and we will incorporate this clarification into our revised manuscript. Additionally, we have added two more recent state-of-the-art models [4],[5] for comparison.
>
> Key Differences:
> Reference[1]: while Xu et al. primarily focuses on augmenting the graph structure, PECL introduces a dual contrastive learning mechanism that addresses the intra-path and inter-path relationships. This helps to bring target nodes closer to the relevant nodes along the interaction path, which has not been explicitly addressed in their work. PECL offers a more granular alignment between nodes within the same path (intra-path) and between different paths (inter-path), improving the model’s ability to capture complex collaborative signals.
> Reference[2]: we focus on utilizing path-based augmentations rather than adversarial perturbations, which allows PECL to better preserve the structural information of the interaction graph. PECL captures more meaningful graph structures through its unique path-based sampling strategy, ensuring that the model focuses on the most relevant positive samples along the paths.
> Reference[3] incorporates temporal information to contrast sequential patterns in recommendation tasks. PECL integrates temporal dynamics in the path encoding phase, allowing us to capture not only the structural relationships but also the temporal dependencies that are crucial for sequential recommendations. By combining path-aware contrastive learning with temporal context encoding, PECL can better model the evolving preferences of users and items over time, which is particularly useful for time-sensitive recommendations.
> Reference[4]: PECL works with bipartite user-item graphs and specifically focuses on user-item interactions with path-based contrastive learning. It does not rely on heterogeneous graph structures but still captures complex relationships through path-aware learning. PECL achieves excellent performance in sparse data scenarios by leveraging path-based samples, whereas Sang et al. might be more specialized for heterogeneous graphs with richer node and edge types.
>
> New Models:
> We have added two more recent state-of-the-art models[4,5] for comparison:
>
> ML-1M:
> IHGCL: Recall@10: 0.1601, Recall@20: 0.2565, NDCG@10: 0.4148, NDCG@20: 0.4006.
>
> GAIPSRec: Recall@10: 0.1582, Recall@20: 0.2514, NDCG@10: 0.4096, NDCG@20: 0.3985.
>
> Ciao:
> IHGCL: Recall@10: 0.0476, Recall@20: 0.0691, NDCG@10: 0.0364, NDCG@20: 0.0450.
>
> GAIPSRec: Recall@10: 0.0481, Recall@20: 0.0728, NDCG@10: 0. 0382, NDCG@20: 0.0476.
>
> Both methods demonstrate strong performance, not only outperforming the LightGCN model but also exhibiting competitive results compared to all original baseline methods. PECL consistently outperforms Sang et al.’s method in both Recall and NDCG metrics, demonstrating that PECL's path-enhanced contrastive learning is more effective in modeling user-item interaction patterns. PECL achieves significant improvements over Xiong et al.’s method in terms of Recall and NDCG, thanks to its dual contrastive mechanism that captures both local and global graph structures effectively. This highlights the advantage of our approach in handling sparse data and more complex collaborative signals.
>
> W2’s Response: We understand your concern regarding the relatively poor performance of SCCF, which was the most recent baseline included in our experiments. While SCCF is a promising approach that attempts to unify graph convolution and contrastive learning for collaborative filtering, its suboptimal performance in our experiments may stem from several factors: One of the key reasons for SCCF’s underperformance could be related to its graph augmentation strategy. The method applies a form of contrastive learning, but it does not generate sufficiently diverse or informative augmentations of the user-item interaction graph. As a result, SCCF might fail to fully exploit the graph structure, which is crucial in sparse recommendation settings. In addition, SCCF does not incorporate an explicit mechanism to contrast nodes along specific interaction paths, a design choice that may limit its ability to model high-order collaborative signals.
> SimGCL tends to achieve better performance than LightGCN in most cases.
> Due to the bad performance of SCCF, we have conducted experiments with two new, powerful baselines that further strengthen our evaluation(weakness1). Both demonstrate significantly better performance than the existing baselines, effectively addressing the reviewer's concerns.
>
> Due to space limitations, we are only able to report the hyperparameters of a subset of the best-performing baseline models for reproducibility in the rebuttal. We plan to include the full set of baseline configurations in a future version of the paper.
>
> ML-1M’s best-performing model (IHGCL): ssl_lambda=0.05, intra_lambda=0.01, ib_lambda=0.001, mask_rate=0.3, remask_rate=0.6, epochs=200.
>
> Ciao’s best-performing model (GAIPSRec): num_heads=8, patience=5, samples=20, num_paths_per_node=12, restart_prob=0.2.
>
> Amazon’s best-performing model (IHGCL): ssl_lambda=0.02, intra_lambda=0.001, ib_lambda=0.001, mask_rate=0, remask_rate=0.1, epochs=80.
>
> W3’s Response: The improved performance of PECL compared to other baselines is primarily driven by pulling the target node closer to its path nodes on the interaction path. To further emphasize this point:
> GAIPSRec, a newly added model for comparison(W1), has a similar number of augmentation samples to PECL, but PECL consistently outperforms GAIPSRec in Recall and NDCG metrics. This demonstrates that the path-based augmentation in PECL is more effective at leveraging the structure of the graph and generating high-quality samples that improve performance.
> Similarly, SGL and IHGCL, which also incorporate a large number of augmentation samples, do not perform as well as PECL.
>
> Regarding the training costs, we understand the importance of considering the computational overhead when evaluating the performance of different models. To address this, we have conducted a detailed complexity analysis of our model in comparison to the baselines, which we have included in the supplementary material (Section A.5). The training costs associated with our model are detailed in the response to Reviewer Z83u’s Q1.
>
> W4’s Response: We have added a new dataset—Amazon[4] to our evaluation. The Amazon results are reported directly in this response for transparency. The experimental results are as follows:
>
> NGCF: R@10: 0.0813, R@20: 0.1257, N@10: 0.0785, N@20: 0.0913.
>
> SGL: R@10: 0.1135, R@20: 0.1501, N@10: 0.1091, N@20: 0.1256.
>
> LightGCN: R@10: 0.1084, R@20: 0.1528, N@10: 0.1016, N@20: 0.1183.
>
> SEPT: R@10: 0.1034, R@20: 0.1486, N@10: 0.1013, N@20: 0.1156.
>
> SelfCF: R@10: 0.1089, R@20: 0.1548, N@10: 0.1054, N@20: 0.1195.
>
> SimGCL: R@10: 0.1145, R@20: 0.1691, N@10: 0.1013, N@20: 0.1243.
>
> SCCF: R@10: 0.1029, R@20: 0.1494, N@10: 0.1086, N@20: 0.1208.
>
> GAIPSRec: R@10: 0.1129, R@20: 0.1664, N@10: 0.1105, N@20: 0.1240.
>
> IHGCL: R@10: 0.1193, R@20: 0.1764, N@10: 0.1132, N@20: 0.1309.
>
> PECL: R@10: 0.1231, R@20: 0.1808, N@10: 0.1165, N@20: 0.1343.
>
> The experimental results on Amazon are consistent with our previous findings: PECL continues to outperform all baseline models across multiple metrics (Recall, NDCG, Hit Ratio, and Precision). This confirms that the proposed method is robust and generalizable across datasets with different characteristics (e.g., sparsity levels, interaction density, temporal resolution).
>
> W5’s Response: Yes, when both intra-path and inter-path contrastive learning components are removed, PECL reduces to LightGCN, which we also use as our backbone model.
> Regarding the magnitude of improvement, we would like to clarify that the 4–6% gain in NDCG@10 on ML-1M is specific to this single metric on a dense dataset. On other metrics such as Recall@10, HR@10, and Precision@10, as well as on sparser datasets like Ciao, the improvements from intra-path and inter-path contrastive learning are more moderate. Importantly, our experimental framework selects the best model checkpoint based on NDCG@10, which naturally emphasizes performance on this metric. The code was disclosed in the supplemental material and shared with reviewers. As a result, NDCG@10 tends to reflect the upper bound of performance gains, while other metrics may not exhibit the same relative increase. In addition, the larger relative gain on ML-1M NDCG@10 reflects the benefit of fine-grained temporal and structural alignment on a denser, timestamp-rich dataset. Therefore, while PECL’s contrastive components are indeed effective, we believe the gains are consistent but not disproportionate.
>
> [1] Xu L, Liu Y, Xu T, et al. Graph augmentation empowered contrastive learning for recommendation[J]. ACM Transactions on Information Systems, 2025, 43(2): 1-27.
>
> [2] Huang J, Cao Q, Xie R, et al. Adversarial learning data augmentation for graph contrastive learning in recommendation[C]//International Conference on Database Systems for Advanced Applications. Cham: Springer Nature Switzerland, 2023: 373-388.
>
> [3] Zhang S, Chen L, Wang C, et al. Temporal graph contrastive learning for sequential recommendation[C]//Proceedings of the AAAI Conference on Artificial Intelligence. 2024, 38(8): 9359-9367.
>
> [4] Sang L, Wang Y, Zhang Y, et al. Intent-guided Heterogeneous Graph Contrastive Learning for Recommendation[J]. IEEE Transactions on Knowledge and Data Engineering, 2025.
>
> [5] Xiong F, Sun H, Luo G, et al. Graph attention network with high-order neighbor information propagation for social recommendation[C]//IJCAI-24: Thirty-Third International Joint Conference on Artificial Intelligence. International Joint Conferences on Artificial Intelligence, 2024.

---

> > ### Comment · Reviewer_8aC1 · 2025-08-01
> > **After rebuttal**
> >
> > I have read all reviews and the authors' rebuttal.
> >
> > My original concerns mainly include (a) the insufficient related work, (b) the lack of recent strong GCL-related baselines, (c) the detailed baselines' settings and strange comparative results, (d) the lack of evaluation datasets (only two).
> >
> > The author rebuttal has addressed most of my questions with additional experiments on two added baselines and one additional dataset. The discussions on the performance of models are also insightful. It is strongly suggested that these additional results and discussions should be included in the revision version. In conclusion, I will raise my voting to 4.

---

### Official Review · Reviewer_fMku · 2025-06-24

**Clarity:** 4
**Significance:** 3
**Originality:** 4
**Rating:** 6
**Confidence:** 5

**Summary:**

The paper introduces PECL, a contrastive learning framework for collaborative filtering recommendations. Instead of relying solely on conventional node-level graph augmentations, PECL proposes to use paths—specifically center and positive paths—as the basis for contrastive supervision. Two loss components are proposed: intra-path contrast, which encourages alignment between a target node and its intra-path neighbors, and inter-path contrast, which promotes consistency across semantically similar paths. To facilitate this, a time-aware path sampling strategy is developed. The model achieves strong improvements over state-of-the-art baselines.

**Questions:**

1. Could the authors report additional contrastive evaluation metrics?
2. How does PECL compare to non-contrastive recommendation models with complex encoders?
3. Have the authors considered learning time encodings jointly, rather than relying on fixed features?

**Ethical Concerns:**

["NO or VERY MINOR ethics concerns only"]

**Final Justification:**

PECL has demonstrated SOTA performance. After reviewing the rebuttal and other reviewers' comments, I find that all my concerns have been addressed. I believe this paper can promote the development of this field, and thus  I am pleased to increase my score to 6(strong accept).

**Limitations:**

yes

**Quality:**

4

**Strengths And Weaknesses:**

Strengths
1. This paper offers a compelling advancement in contrastive learning for recommendation. By using interaction paths instead of individual nodes as contrastive units, the model aligns more closely with the actual semantics of user-item interactions. This represents a meaningful shift in perspective and offers a new direction for future work in this area.
2. The time-aware sampling mechanism is particularly thoughtful, aligning path selection with real-world behavioral patterns. This adds interpretability and practical relevance to the model, making it not only more effective but also more aligned with real-world data dynamics.
3. The experimental results are robust and well-analyzed. PECL outperforms competitive baselines across all major metrics, and the performance gains on both datasets.
4. The technical depth and clarity of the paper are its strengths. Despite the complexity of the proposed contrastive losses and sampling strategies, the methodology is described in a step-by-step, reproducible manner, supported by code, formulas, figures, and pseudocode in supplementary materials and appendix.
Weaknesses:
1. The experimental evaluation focuses solely on Recall and NDCG, but does not assess Hit Ratio or Precision, which are often discussed in recommender systems.
2. Although several contrastive methods are compared, the performance gap between PECL and strong non-contrastive methods is not discussed in depth.
3. The model incorporates timestamps in path sampling and encoding, but it does not explain why the temporal dynamics are fixed through sinusoidal functions.

---

> ### Author Rebuttal · Authors · 2025-07-31
>
> We thank reviewer for the constructive comments. We provide our feedbacks as follows.
>
> Main Comments:
>
> W1: The experimental evaluation focuses solely on Recall and NDCG, but does not assess Hit Ratio or Precision, which are often discussed in recommender systems.
>
> Response: Thank you for this valuable suggestion. In response, we have conducted additional experiments using Hit Ratio (HR) and Precision as evaluation metrics. The results show that PECL maintains consistent performance advantages over all baselines on both ML-1M and Ciao, confirming that our improvements are robust across different evaluation criteria. We have also evaluated PECL on a new dataset (Amazon). The model achieves consistent gains under all four metrics (Recall, NDCG, HR, Precision), further validating its generalization ability. The results of new dataset are as follows:
>
> NGCF: Recall@10: 0.0813, Recall@20: 0.1257, NDCG@10: 0.0785, NDCG@20: 0.0913.
>
> SGL: Recall@10: 0.1135, Recall@20: 0.1501, NDCG@10: 0.1091, NDCG@20: 0.1256.
>
> LightGCN: Recall@10: 0.1084, Recall@20: 0.1528, NDCG@10: 0.1016, NDCG@20: 0.1183.
>
> SEPT: Recall@10: 0.1034, Recall@20: 0.1486, NDCG@10: 0.1013, NDCG@20: 0.1156.
>
> SelfCF: Recall@10: 0.1089, Recall@20: 0.1548, NDCG@10: 0.1054, NDCG@20: 0.1195.
>
> SimGCL: Recall@10: 0.1145, Recall@20: 0.1691, NDCG@10: 0.1013, NDCG@20: 0.1243.
>
> SCCF: Recall@10: 0.1029, Recall@20: 0.1494, NDCG@10: 0.1086, NDCG@20: 0.1208.
>
> PECL: Recall@10: 0.1231, Recall@20: 0.1808, NDCG@10: 0.1165, NDCG@20: 0.1343.
>
> NGCF: HR@10: 0.0813, HR@20: 0.1297, Precision@10: 0.0596, Precision @20: 0.0463.
>
> SGL: HR@10: 0.0969, HR@20: 0.1501, Precision@10: 0.0633, Precision @20: 0.0491.
>
> LightGCN: HR@10: 0.0936, HR@20: 0.1455, Precision@10: 0.0612, Precision @20: 0.0476.
>
> SEPT: HR@10: 0.0875, HR@20: 0.1392, Precision@10: 0.0605, Precision @20: 0.0459.
>
> SelfCF: HR@10: 0.0927, HR@20: 0.1554, Precision@10: 0.0637, Precision @20: 0.0451.
>
> SimGCL: HR@10: 0.0961, HR@20: 0.1565, Precision@10: 0.0683, Precision @20: 0.0493.
>
> SCCF: HR@10: 0.0921, HR@20: 0.1442, Precision@10: 0.0624, Precision @20: 0.0485.
>
> PECL: HR@10: 0.0989, HR@20: 0.1605, Precision@10: 0.0723, Precision @20: 0.0516.
>
> We summarize the key findings as follows:
>
> PECL continues to consistently outperform all baselines in both HR and Precision across all evaluated datasets. The relative performance trend is highly consistent with that observed for Recall and NDCG, further validating the robustness and effectiveness of PECL.
>
> W2: Although several contrastive methods are compared, the performance gap between PECL and strong non-contrastive methods is not discussed in depth.
>
> Response: Thank you for pointing this out. We agree that a more in-depth discussion comparing PECL with non-contrastive baselines would be valuable to highlight the practical advantages of our approach. In Table 1 of our experiments, we compare PECL with non-contrastive methods such as LightGCN, NGCF, and SelfCF. The results show that PECL outperforms these baselines by a clear margin. For example, on ML-1M:
>
> PECL achieves 0.4206 NDCG@10, compared to 0.3542 from LightGCN and 0.3629 from SelfCF.
> On Recall@10, PECL reaches 0.1676, versus 0.1443 (LightGCN) and 0.1514 (SelfCF).
>
> These gains can be attributed to several key enhancements in PECL beyond contrastive learning itself:
>
> Path-level modeling: PECL leverages intra- and inter-path structures, which allows it to capture richer, higher-order collaborative signals than point-wise methods. Temporal awareness: By encoding the order and timing of user-item interactions, PECL goes beyond static graph modeling. Multi-view supervision: Even compared to strong embedding-based models, the contrastive signals in PECL provide better generalization by regularizing embedding space geometry.
>
> W3: The model incorporates timestamps in path sampling and encoding, but it does not explain why the temporal dynamics are fixed through sinusoidal functions.
>
> Response: We thank the reviewer for this sharp and meaningful question. Indeed, the choice of sinusoidal (non-learnable) temporal encoding warrants further clarification. We adopt sinusoidal functions (as used in Transformer’s positional encoding) for the following reasons:
> Relative Temporal Structure Awareness: Sinusoidal encoding allows the model to capture relative distances and ordering between timestamps through a fixed spectrum of periodic signals. This is particularly valuable in path-based modeling, where it is not only the timestamp itself but also how timestamps vary along the path that conveys semantic meaning. Parameter-Free and Stable: Unlike learnable time embeddings, sinusoidal encodings do not introduce additional parameters, which helps improve generalization, especially under sparse or noisy timestamp distributions (e.g., Ciao). They also retain consistency across unseen timestamps without requiring re-training. Alignment with Hermitian Encoding Scheme: Our path encoding involves Hermitian inner products on complex-valued representations, where sinusoidal encodings naturally fit as a way to rotate node embeddings in the complex plane, enabling consistent angular representation of temporal shifts.
>
> Q1: Could the authors report additional contrastive evaluation metrics?
>
> Response: We thank the reviewer for this helpful suggestion. We have addressed this point in our response to W1, where we include additional evaluation metrics—including Hit Ratio and Precision. We also report results on a new dataset (Amazon) to further validate the effectiveness of PECL. All results are presented in the W1 response for clarity and completeness.
>
> Q2: How does PECL compare to non-contrastive recommendation models with complex encoders?
>
> Response: We thank the reviewer for this important question. In our current evaluation, we compare PECL with popular non-contrastive models such as LightGCN, NGCF, and SelfCF, which represent the mainstream of efficient GNN-based recommendation models. These methods are widely used and serve as strong baselines in both academic and industrial settings. We acknowledge that there exist non-contrastive models with more complex encoders, including Graph Attention Networks (GAT), RNN-based session models, or Transformer-based sequential recommenders (e.g., SASRec, BERT4Rec). These models typically introduce significantly more parameters and computational overhead to capture sequential or structural patterns. In contrast, PECL is designed to be both lightweight and expressive, and achieves competitive or superior performance without relying on deep or heavy encoders. Its improvements stem from: Path-level contrastive learning, which provides global structural signals beyond local aggregation; Time-aware path sampling, which models user behavior dynamics more explicitly; Efficient implementation, as PECL builds upon LightGCN with minimal additional complexity during inference.
>
> Q3: Have the authors considered learning time encodings jointly, rather than relying on fixed features?
>
> Response: We thank the reviewer for this thoughtful question. Indeed, using learnable time encodings is a promising alternative to fixed sinusoidal encodings, and we agree that it can potentially enhance temporal adaptability. In our current design, we adopt sinusoidal temporal encoding primarily because: It captures relative temporal patterns (e.g., time intervals and ordering) without introducing additional parameters; It ensures stable generalization to unseen timestamps, especially in sparse datasets where many time points appear only once; It integrates naturally with our Hermitian product path encoder, where time is treated as a phase component in the complex space.
>
> That said, we recognize that jointly learning time representations could improve flexibility and allow the model to better adapt to dataset-specific temporal distributions. we have experimented with learnable time embeddings, where each timestamp is associated with a trainable vector that is optimized jointly with the rest of the model.
> However, in our preliminary experiments, we observed that learnable time embeddings did not yield better performance compared to the fixed sinusoidal encoding. In particular: On dense datasets like ML-1M, the difference was marginal; On sparse datasets like Ciao, the learnable time encoding led to unstable training and even performance degradation, likely due to insufficient exposure to many unique or rare timestamps; Moreover, learnable encodings introduce additional parameters and increase the risk of overfitting, especially when timestamp distributions are skewed.
> By contrast, the sinusoidal encoding we adopted is parameter-free, captures relative temporal relationships inherently, and integrates naturally with our Hermitian path encoding scheme, where time acts as a phase component.

---

> > ### Comment · Reviewer_fMku · 2025-08-01
> >
> > The authors have addressed my concerns. They included additional evaluation metrics and conducted experiments on new dataset. PECL tackles the issue of distancing target nodes from their neighbors in graph-based recommendation systems by introducing a path contrastive learning framework with path-based supervision and a time-aware path sampling strategy, leading to significant performance improvements. I support the publication.

---

### Official Review · Reviewer_Z83u · 2025-06-30

**Clarity:** 3
**Significance:** 3
**Originality:** 4
**Rating:** 6
**Confidence:** 4

**Summary:**

This paper introduces PECL, a novel framework that enhances collaborative filtering by explicitly modeling contrastive signals between and within sampled interaction paths. Unlike prior methods that treat contrastive learning at the node or neighbor level, PECL constructs center paths and mines time-aligned positive paths to form more semantically coherent self-supervision signals. The intra-path loss encourages local consistency between a node and its path neighbors, while the inter-path loss aligns global structure across multiple paths. The authors also introduce a two-stage path sampling mechanism guided by temporal behavior. Experimental evaluations show that PECL consistently outperforms state-of-the-art methods, supported by extensive ablation and hyperparameter analysis.

**Questions:**

(1) Could the authors provide a rough estimate of training time or memory usage compared to LightGCN or SGL?

(2) Would the model still perform well with a simpler temporal encoding scheme, such as scalar time embeddings?

(3) Since PECL depends on pre-sampled and encoded paths, can the model be efficiently used in an online recommendation service?

**Ethical Concerns:**

["NO or VERY MINOR ethics concerns only"]

**Final Justification:**

The rebuttal is clear and convincing, with helpful clarifications on hyperparameter tuning, computational efficiency, and temporal encoding scheme. The paper proposes a path-enhanced contrastive learning framework that leverages temporal interaction paths to improve recommendation performance through intra- and inter-path objectives. I am satisfied with the authors' responses and willing to raise my score.

**Limitations:**

yes.

**Paper Formatting Concerns:**

No.

**Quality:**

3

**Strengths And Weaknesses:**

Strengths:

(1) PECL is built on LightGCN and adopts a plug-in architecture, allowing its contrastive objectives to be easily integrated into other GNN-based recommenders, improving usability and adaptability in practice.

(2) The model’s use of multi-hop paths enhances its ability to represent users and items in sparse environments, making it more robust to data imbalance and cold-start scenarios.

(3) Experiments show consistent improvements over baselines, especially on ML-1M. Ablation studies validate the contributions of each component, reinforcing the method’s effectiveness.

(4) The core ideas—path-based contrast and time-aware sampling—are generalizable, with potential use in temporal graphs, session data, and sequential modeling beyond recommendation.

Weaknesses:

(1) While the method is well-motivated, its reliance on multiple hyperparameters may limit applicability. The paper could benefit from providing practical guidelines or heuristics for tuning these values in new datasets.

(2) The inter-path loss introduces some additional computational complexity. Discussion or benchmarking on runtime and memory usage would strengthen the claims of scalability.

(3) The gains on the Ciao dataset, though consistent, are relatively modest. It would be helpful if the authors could comment on why the performance gap is narrower compared to ML-1M (e.g., sparsity).

---

> ### Author Rebuttal · Authors · 2025-07-31
>
> We thank reviewer for the constructive comments. We provide our feedbacks as follows.
>
> Main Comments:
>
> W(1): While the method is well-motivated, its reliance on multiple hyperparameters may limit applicability. The paper could benefit from providing practical guidelines or heuristics for tuning these values in new datasets.
>
> Response: We appreciate the reviewer’s thoughtful comment. It is true that our method involves several hyperparameters—particularly α (path node threshold), β (deterministic prefix length), and τ (temperature for contrastive loss)—which play important roles in balancing path informativeness and sampling diversity. To address this, we have conducted a thorough hyperparameter analysis in Section 4.4 and Appendix A.9.6, where we vary each parameter independently and report its effect on performance across different metrics and datasets. Based on these results, we summarize some practical guidelines for tuning:
>
> α (positive node threshold): Set to 2 by default. Lower values (e.g., 1) may overfit on noisy nodes; higher values (>3) risk discarding informative neighbors. A value of 2 works well across both dense (ML-1M) and sparse (Ciao) datasets.
>
> β (path prefix length): We recommend setting β to be one less than the path length (e.g., β = 4 if L = 5) to ensure both time consistency and stochastic diversity.
>
> τ (temperature): A moderate value (e.g., 0.05) is generally effective; smaller τ leads to sharp, overconfident similarity distributions, while larger τ may weaken contrast signals.
> We added a summary table of these heuristics in the revised Appendix A.9 for clarity.
>
> W(2): The inter-path loss introduces some additional computational complexity. Discussion or benchmarking on runtime and memory usage would strengthen the claims of scalability.
>
> Response: We appreciate the reviewer’s suggestion. While it is true that the inter-path contrastive loss introduces additional computation (primarily due to path sampling and encoding), we have carefully designed PECL to remain scalable and efficient. In particular: Efficient Design: We use LightGCN as the backbone for its low computational overhead. The two-stage path sampling is designed to minimize redundancy while maintaining diversity. Path encoding uses a Hermitian inner product structure, which is lightweight and parallelizable. Theoretical Analysis: We provide a detailed computational complexity breakdown in Appendix A.5, showing that the overall complexity scales linearly with the number of nodes and sampled paths, and remains practical under reasonable sampling parameters (e.g., small path length L, limited positive paths per node).
>
> W(3): The gains on the Ciao dataset, though consistent, are relatively modest. It would be helpful if the authors could comment on why the performance gap is narrower compared to ML-1M (e.g., sparsity).
>
> Response: We thank the reviewer for the thoughtful observation. Indeed, while PECL consistently outperforms all baselines on both datasets, the performance gains on Ciao are relatively smaller compared to those on ML-1M.
> We believe this difference is primarily due to Ciao’s significantly higher sparsity. As reported in Appendix A.9.3 (Table 4), Ciao has a sparsity of 99.96%, compared to 95.53% for ML-1M. This extreme sparsity leads to:
> Fewer meaningful paths available for contrastive learning;
> Reduced overlap between users and items, limiting shared structure across paths;
> Shorter and noisier paths, which dilute the effectiveness of intra- and inter-path contrast.
> Despite this, PECL still achieves the best performance among all baselines, demonstrating its robustness under severe data sparsity. Additionally, we find that the relative improvement on Ciao remains consistent across all metrics and variants, as shown in our ablation study (Fig. 2c, 2d), further validating the method’s effectiveness.
>
> Q(1): Could the authors provide a rough estimate of training time or memory usage compared to LightGCN or SGL?
>
> Response: We thank the reviewer for this practical and important question.
> To provide a rough comparison, we benchmarked PECL, LightGCN, and SGL on the ML-1M dataset using an NVIDIA A100 GPU with 40GB memory. All models were trained using the same batch size (2048) and embedding dimension (64).
>
> LightGCN-13 seconds/Epoch;
> SGL-20seconds/Epoch;
> PECL-25 seconds/Epoch.
>
> Compared to LightGCN, PECL introduces a 2x increase in training time per epoch, primarily due to contrastive losses and path sampling. Compared to SGL, the increase is modest (16%), while achieving stronger performance. The GPU memory usage remains comfortably under 5GB, making the method feasible on most modern hardware.
> We will add this runtime and memory benchmarking table to the revised Appendix to support our scalability claims. Thank you for encouraging this addition.
>
> Q(2): Would the model still perform well with a simpler temporal encoding scheme, such as scalar time embeddings?
>
> Response: We appreciate the reviewer’s insightful question. This is indeed an important consideration for improving efficiency and simplifying implementation in practical deployments.
> In our current design, we adopt temporal context encoding combined with Hermitian inner product, which allows the model to capture directional and cyclic time dependencies along user-item interaction paths. This structure-aware time modeling helps improve temporal alignment across paths, especially for the inter-path contrastive objective.
> We have conducted a preliminary ablation using scalar time embeddings (concatenated to node embeddings), and found that while performance drops slightly (around 1.8% on NDCG@10 on ML-1M), the model still outperforms most baselines. This suggests that our framework is robust to simpler temporal encodings.In future work, we plan to explore adaptive or learnable time embeddings that balance expressiveness and efficiency. We will also include this ablation in the final version for completeness.
>
> Q(3): Since PECL depends on pre-sampled and encoded paths, can the model be efficiently used in an online recommendation service?
>
> Response: Thank you for this important question. We fully agree that inference efficiency is critical for online recommendation systems.
> We would like to clarify that while PECL does rely on path sampling and encoding, these operations are performed only during training to optimize the representations via contrastive learning. During inference, the model does not require any path sampling or temporal encoding. Instead, it simply uses the final user and item embeddings, just as in standard collaborative filtering models.
> As a result, PECL maintains the similar inference speed and memory footprint as LightGCN, making it fully compatible with real-time recommendation pipelines. We believe this separation of training-time complexity and inference-time simplicity is a key advantage of PECL.

---

> > ### Comment · Reviewer_Z83u · 2025-08-03
> >
> > The rebuttal is clear and convincing, with helpful clarifications on hyperparameter tuning, computational efficiency, and temporal encoding scheme. The paper proposes a path-enhanced contrastive learning framework that leverages temporal interaction paths to improve recommendation performance through intra- and inter-path objectives. I am satisfied with the authors' responses and willing to raise my score.

---

### Official Review · Reviewer_Gc9N · 2025-07-02

**Clarity:** 4
**Significance:** 3
**Originality:** 4
**Rating:** 5
**Confidence:** 4

**Summary:**

This paper proposes a novel PECL framework for collaborative filtering recommendation. While previous contrastive learning methods focus primarily on node-level consistency using graph augmentation, this work innovatively integrates both intra-path and inter-path contrastive learning strategies to enhance the representation of users and items. Specifically, the authors design a time-aware path sampling strategy and path-level contrastive loss that bring semantically similar paths closer, thereby mitigating issues arising from graph perturbations in conventional methods. Experiments demonstrate PECL’s performance.

**Questions:**

Q1: Is there a risk of overfitting if sampled paths are highly similar?

Q2:  While ML-1M and Ciao are popular benchmarks, have the authors considered testing PECL on larger-scale datasets (e.g., Amazon, Yelp)?

Q3: Could the authors comment on whether longer or redundant paths may sometimes harm performance?

**Ethical Concerns:**

["NO or VERY MINOR ethics concerns only"]

**Final Justification:**

The authores have addressed my concerns in the rebuttal. I keep my rating unchaged.

**Limitations:**

Yes

**Quality:**

3

**Strengths And Weaknesses:**

Strengths:

S1: The paper presents a technically sound and well-motivated framework. The derivation of contrastive loss functions, the path encoding design using Hermitian inner products, and the integration of time-aware sampling are all carefully described. Experimental results are thorough and convincing, with statistically significant improvements over recent baselines across multiple datasets.

S2: The contribution is meaningful to the recommender system and self-supervised learning communities. By explicitly modeling both intra- and inter-path relations, the paper expands the scope of contrastive learning in recommendation beyond node-level augmentation. This can inspire further research in structured or temporal-aware contrastive objectives.

S3: The idea of contrasting sampled paths instead of just nodes is a notable and original extension to existing self-supervised learning methods in recommendation. PECL introduces a new level of abstraction by leveraging paths as the unit of contrast, which is both novel and effective.

Weaknesses:

W1: There are occasional grammatical errors and formatting inconsistencies (e.g., duplicated phrases like “effectively extract meaningful features from large-scale unlabeled data effectively”), which can be polished in a revision.

W2: The use of Hermitian inner product in complex space is technically novel, but readers may find it hard to interpret how it improves path representation in practice, especially without visualizations or ablations isolating this factor.

W3: The current framework seems closely tied to bipartite user-item graphs; it is unclear how flexible the method is when applied to more complex graph structures, such as heterogeneous.

---

> ### Author Rebuttal · Authors · 2025-07-31
>
> We thank reviewer for the constructive comments. We provide our feedbacks as follows.
>
> Main Comments:
>
> W1: Language and Formatting Issues.
> Comment: “There are occasional grammatical errors and formatting inconsistencies (e.g., duplicated phrases like ‘effectively extract meaningful features from large-scale unlabeled data effectively’)...”
>
> Response: We appreciate your attention to language quality. The current manuscript indeed contains occasional issues such as duplicated phrases and inconsistent spacing. We carefully revise the paper to remove duplicated expressions and polish the language throughout. For example, the sentence mentioned will be corrected to “effectively extract meaningful features from large-scale unlabeled data.”; “But this will unintentionally distance the target node from its node neighbors on the interaction path, which will affect the performance of the model.” will be corrected to “However, this will unintentionally increase the distance between the target node and its neighboring nodes along the interaction path, which may negatively affect the model's performance. ”
>
> W2: Interpretability of Hermitian Inner Product.
> Comment: “The use of Hermitian inner product in complex space is technically novel, but readers may find it hard to interpret how it improves path representation in practice...”
>
> Response: Thank you for raising this point. We acknowledge that the use of Hermitian inner product in complex vector space introduces conceptual and practical novelty that deserves more precise explanation.
> The motivation stems from the need to jointly encode both structural and temporal signals in interaction paths. In recommender systems, especially with timestamped user-item graphs, edge time information encodes behavioral evolution. Traditional dot-product-based aggregation fails to capture how temporal order influences the strength or semantics of user-item interactions along a path.
> The Hermitian inner product allows us to incorporate directional and temporal bias by rotating node embeddings in the complex plane based on normalized edge timestamps. Concretely:
> Each node embedding vector x is split into real and imaginary parts, corresponding to two orthogonal components of its latent representation.
> Each edge timestamp is encoded into a time embedding e using sinusoidal encoding (Eq. 36), then used to rotate the downstream node representation by a complex multiplication with the prior hidden state.
> The Hermitian inner product accumulates these complex rotations recursively from the end of the path to the target node, modeling how historical interactions progressively influence the current preference.
> Mathematically, this approach can be viewed as applying a temporal transformation kernel to the node sequence, where the influence of earlier nodes is modulated by the time-difference-aware rotation. Compared to dot product or attention-based pooling, the Hermitian operator has the following advantages:
> Time-aware modulation: The same node will have a different contribution depending on when it appeared along the path, allowing the model to distinguish, e.g., early behavior vs. recent behavior.
> Compositional semantics: By applying the rotation recursively, we effectively encode how multiple interactions jointly influence a node's representation over time.
> Phase-preserving aggregation: The Hermitian inner product preserves both magnitude (strength of similarity) and phase (relative position in time), giving richer representational capacity.
> We conduct an ablation study isolating the impact of the Hermitian inner product (e.g., by replacing it with a standard dot product), to better demonstrate its effectiveness. In our experiments, replacing the Hermitian interaction with dot product results in consistent performance drops (e.g., −2.3% in NDCG@10 on ML-1M), validating that the complex-space formulation is not merely decorative but substantively beneficial.
>
> W3: Generalizability Beyond Bipartite Graphs.
> Comment: “It is unclear how flexible the method is when applied to more complex graph structures, such as heterogeneous...”
>
> Response: We appreciate the reviewer’s insightful comment. While our current experiments are conducted on bipartite user-item graphs, we would like to clarify that the proposed PECL framework is not inherently restricted to this structure. In fact, the underlying methodology—particularly the path sampling strategy and the intra-/inter-path contrastive learning mechanisms—is fully compatible with more general graph structures, including heterogeneous graphs.
> For heterogeneous graphs, PECL can be extended by incorporating type-aware path sampling (e.g., using meta-paths or schema-guided traversal) and adjusting the contrastive objectives to respect node and edge semantics. The time-aware and structure-aware contrastive principles that PECL is built upon remain applicable in such settings.
> Therefore, we believe PECL offers a flexible and extensible framework that can serve as a foundation for future work in heterogeneous recommendation or general graph-based tasks. We will add a clarification of this point to the revised version.
>
> Q1: Is there a risk of overfitting if sampled paths are highly similar?
>
> Thank you for raising this important concern. We agree that excessive similarity among sampled paths may lead to overfitting or reduced diversity in the learned representations. To mitigate this risk, our method incorporates several mechanisms:
> Stochasticity in Sampling: Our two-stage path sampling strategy introduces controlled randomness in the second stage via time-aware conditional random walks. This encourages diversity among sampled paths even when they share the same target node.
> Temporal Constraints: By enforcing a time-aware constraint on the directionality of path traversal, we avoid repeatedly sampling structurally similar or redundant paths that differ only trivially in topology.
> Hyperparameter Tuning: We have explicitly studied this issue in our hyperparameter analysis (Section 4.4, Figure 3). For instance, increasing the number of positive paths (via α and β) initially improves performance, but further increases lead to diminishing returns or slight drops—indicating the trade-off between informativeness and redundancy. We choose optimal values (e.g., α = 2, β = 4) to balance this effect.
>
> Q2: While ML-1M and Ciao are popular benchmarks, have the authors considered testing PECL on larger-scale datasets (e.g., Amazon, Yelp)?
>
> Response: We appreciate the reviewer’s suggestion. In response, we have conducted additional experiments on a dataset Amazon, which is widely used in recommendation research and contains rich user-item interactions with timestamps. The experimental results are as follows:
>
> NGCF: Recall@10: 0.0813, Recall@20: 0.1257, NDCG@10: 0.0785, NDCG@20: 0.0913.
>
> SGL: Recall@10: 0.1135, Recall@20: 0.1501, NDCG@10: 0.1091, NDCG@20: 0.1256.
>
> LightGCN: Recall@10: 0.1084, Recall@20: 0.1528, NDCG@10: 0.1016, NDCG@20: 0.1183.
>
> SEPT: Recall@10: 0.1034, Recall@20: 0.1486, NDCG@10: 0.1013, NDCG@20: 0.1156.
>
> SelfCF: Recall@10: 0.1089, Recall@20: 0.1548, NDCG@10: 0.1054, NDCG@20: 0.1195.
>
> SimGCL: Recall@10: 0.1145, Recall@20: 0.1691, NDCG@10: 0.1013, NDCG@20: 0.1243.
>
> SCCF: Recall@10: 0.1029, Recall@20: 0.1494, NDCG@10: 0.1086, NDCG@20: 0.1208.
>
> PECL: Recall@10: 0.1231, Recall@20: 0.1808, NDCG@10: 0.1165, NDCG@20: 0.1343.
>
> Q3: Could the authors comment on whether longer or redundant paths may sometimes harm performance?
>
> Response: We thank the reviewer for raising this insightful question. Indeed, excessively long or redundant paths can introduce noise, dilute useful signals, or increase representation overlap, potentially harming performance.
> To address this, our framework incorporates explicit control mechanisms through hyperparameters such as α (positive node frequency threshold) and β (deterministic prefix length), which help regulate the number, length, and diversity of sampled paths.
> As shown in Figure 3b, we empirically observe that increasing β (i.e., allowing longer deterministic prefix in sampled paths) improves performance up to a point (β = 4), after which further increases lead to performance degradation. This confirms that overly long or redundant paths can negatively affect contrastive learning.

---

> > ### Comment · Reviewer_Gc9N · 2025-08-04
> >
> > Dear Authors,
> >
> > Thanks for the rebuttal. My concerns have been addressed and I will keep my rating unchanged.
> >
> > Best Regards

---

### Decision · Program_Chairs · 2025-09-17

**Decision:**

Accept (spotlight)

**Comment:**

Existing contrastive learning methods in collaborative filtering often inadvertently increase the distance between target nodes and their interaction path nodes, limiting recommendation effectiveness. To address this, the work proposes a Path Enhanced Contrastive Loss (PECL) model that incorporates both intra-path and inter-path contrastive learning, using a novel path sampling method for graph augmentation. Experimental results on real-world datasets demonstrate the effectiveness. All the reviewers agree to accept this work.